# Adaptive gene loss in the common bean pan-genome during range expansion and domestication

Gaia Cortinovis[1,15], Leonardo Vincenzi [2,15], Robyn Anderson [3], Giovanni Marturano[2], Jacob Ian Marsh[3], Philipp Emanuel Bayer [3], Lorenzo Rocchetti[1], Giulia Frascarelli[1], Giovanna Lanzavecchia[1], Alice Pieri[1], Andrea Benazzo[4], Elisa Bellucci [1], Valerio Di Vittori [1], Laura Nanni [1], Juan José Ferreira Fernández [5], Marzia Rossato [2,6], Orlando Mario Aguilar[7], Peter Laurent Morrell [8], Monica Rodriguez [9,10], Tania Gioia [11], Kerstin Neumann[12], Juan Camilo Alvarez Diaz[13], Ariane Gratias[13], Christophe Klopp [14], Elena Bitocchi [1], Valérie Geffroy[13] ✉, Massimo Delledonne [2,6] ✉, David Edwards [3] ✉ & Roberto Papa [1] ✉

The common bean (*Phaseolus vulgaris* L.) is a crucial legume crop and an ideal evolutionary model to study adaptive diversity in wild and domesticated populations. Here, we present a common bean pan-genome based on five high-quality genomes and whole-genome reads representing 339 genotypes. It reveals ~234 Mb of additional sequences containing 6,905 protein-coding genes missing from the reference, constituting 49% of all presence/absence variants (PAVs). More non-synonymous mutations are found in PAVs than core genes, probably reflecting the lower effective population size of PAVs and fitness advantages due to the purging effect of gene loss. Our results suggest pan-genome shrinkage occurred during wild range expansion. Selection signatures provide evidence that partial or complete gene loss was a key adaptive genetic change in common bean populations with major implications for plant adaptation. The pan-genome is a valuable resource for food legume research and breeding for climate change mitigation and sustainable agriculture.

Food legumes provide valuable resources to address global challenges such as climate change, biodiversity conservation, and the need for sustainable agriculture and healthy diets[1–3]. The common bean (*Phaseolus vulgaris* L.) is a diploid ($2n = 2x = 22$) and predominantly self-pollinating annual grain legume crop with a prominent role in agriculture and society[4–6]. It is also an ideal evolutionary model to study adaptive diversity in wild and domesticated legume populations[7].

The use of *P. vulgaris* as an evolutionary model reflects the parallel and independent life history of two geographically isolated, genetically differentiated gene pools (Mesoamerican and Andean) following its

wild expansion from Mexico to South America ~150,000–200,000 years ago, long before its dual domestication[8–11]. Previous studies using a single reference genome have provided insights into the population structure of the common bean[12] and the genetic basis of important adaptive traits[13]. However, pan-genomic diversity must be explored in detail to gain a more comprehensive understanding[14–17].

Here, we describe the construction of a *P. vulgaris* pan-genome using a non-iterative approach and an analysis of its genetic diversity in terms of presence/absence variants (PAVs) within a representative panel of genetically and phenotypically well-characterized

A full list of affiliations appears at the end of the paper. ✉e-mail: valerie.geffroy@universite-paris-saclay.fr; massimo.delledonne@univr.it; Dave.Edwards@uwa.edu.au; r.papa@univpm.it

accessions. This publicly available common bean pan-genome will provide a valuable starting point to identify genes and genomic mechanisms affecting adaptation, and will accelerate the improvement of food legume crops.

## Results and discussion
### Characterization of the common bean pan-genome
To generate the common bean pan-genome, we applied a non-iterative approach to five high-quality de novo genome assemblies of wild and domesticated genotypes and incorporated short-read whole genome sequencing (WGS) data from 339 representative common bean accessions, comprising 33 wild and 306 domesticated forms. This revealed ~234 Mb of additional sequence containing 6905 genes missing from the reference genome. These regions, termed non-reference regions (NRRs), expand our comprehension of common bean diversity. Indeed, these sequences account for 20% of the total pan-genes, with 7.5% (2579 genes) derived from the high-quality genomes and the remaining 12.5% (4326 genes) from the panel of 339 WGS genotypes. The final size of the reconstructed pan-genome was ~770 Mb, with 34,338 predicted protein-coding genes (Supplementary Tables 1 and 2).

The reference pan-genome was used for variant and PAV calling (Supplementary Data 1). We detected 23,343,365 variant sites, made up of 19,002,047 single-nucleotide variants (SNVs) and 4,341,318 insertions/deletions (InDels). Following PAV calling, the categorization of all 34,338 predicted genes by frequency revealed that 59% of the pan-genome consists of core genes present across all lines (20,369 genes), with the remaining 41% comprising 13,969 PAVs encompassing genes partially shared among accessions or private to a single genotype. Notably, 49% of these PAVs (6905 genes) originate from NRRs (Supplementary Table 2). The growth curve related to the size calculation suggested a closed pan-genome. In agreement, the pan-genes reached the saturation point (99%, 33,997 genes) and remained constant without substantial increases when the number of accession genomes exceeded 125. In contrast, the size of the core gene set decreased with each added genotype (Fig. 1a). This indicates that the final pan-genome includes almost all the gene content of *P. vulgaris*. Gene Ontology (GO) enrichment analysis showed that the core genes are enriched for terms associated with homeostatic (GO:0042592) and catabolic (GO:0043632) processes (Supplementary Fig. 1 and Supplementary Data 2) whereas the PAVs are enriched for terms related to defense (GO:0006952), responses to external stimuli (GO:0009605), responses to light (GO:0019684), and reproduction (GO:0000003, GO:0022414) (Supplementary Fig. 2 and Supplementary Data 3).

To investigate the evolution of the core genes and PAVs, we calculated the non-synonymous and synonymous ratio (Ka/Ks) for each gene in each accession (Supplementary Data 4). This revealed a statistically significant difference ($p < 2.2 \times 10^{-16}$), with PAVs exhibiting a higher Ka/Ks ratio compared to core genes (Supplementary Fig. 3). When we split the PAVs into three subcategories based on their frequency (soft-core $0.90 \leq$ freq. $< 1$; accessory $0.10 \leq$ freq. $< 0.90$; and rare freq. $< 0.10$), we observed a significant increase ($p = 0.03$) in the Ka/Ks ratio among the rare genes compared to the soft-core genes (Fig. 1b and Supplementary Table 3). These results may reflect the lower effective population size of the PAVs (reducing the efficiency of purifying selection) and the higher fitness gain from purging genes that have accumulated non-synonymous (loss-of-function) mutations.

### Evolutionary trajectory of the common bean
The common bean is characterized by three eco-geographic gene pools. Mesoamerican (M) and Andean (A) populations, which encompass both wild and domesticated forms, constitute most of the species diversity, while a third originates from Northern Peru/Ecuador (PhI) and has a relatively narrow distribution of only wild individuals[11]. The Mesoamerican and Andean gene pools include five domesticated subgroups (M1, M2, A1, A2 and A3) corresponding to the Durango-Jalisco, Mesoamerica, Nueva Granada, Peru, and Chile races[13]. We constructed neighbor-joining (NJ) phylogenetic trees (Fig. 2a and Supplementary Fig. 4) and conducted PAV-based principal component analysis (PCA) (Fig. 2b), both of which confirmed this well-defined population structure. Both analyses further divided the M1/Durango-Jalisco races into clusters that we named A and B, respectively. The analysis of variance conducted on M1/Durango-Jalisco accessions, considering the first component for the days to flowering (PC1_DTF)[13], revealed that cluster A flowers significantly later than cluster B ($p < 0.0001$; Fig. 2c and Supplementary Table 4). This genetically distinguished the M1/Durango-Jalisco races in relation to a key adaptive trait (flowering time), indicating that the use of the pan-genome as a reference enhances the characterization of the genetic diversity present in *P. vulgaris* and consequently improves its analysis, exploitation, and management. Cumulatively, the first and the second principal components of the PAV-based PCA explained 46.6% of the total variance, where PC1 mainly defined the differences between the Mesoamerican and Andean gene pools while PC2 split the groups and subgroups within each gene pool (Fig. 2b). The NJ trees further underscored the greater suitability of core genes rather than PAVs for phylogenetic reconstruction because they mitigate biases arising from the absence of genetic material among compared accessions. In contrast to the tree based on single-nucleotide polymorphisms (SNPs) located on PAVs (Supplementary Fig. 4), the NJ tree based solely on core SNPs properly grouped the wild PhI accession close to the wild Mesoamerican genotypes originating from Guatemala and Costa Rica (Fig. 2a), which are most closely related to the PhI gene pool[11].

When we examined the total number of PAVs per genetic group (Supplementary Table 5), we found that wild Mesoamerican and Andean populations have a greater number of genes compared to their domesticated counterparts (Fig. 2d). This supports the well-established notion that domestication is usually associated with a reduction of genetic diversity. Indeed, the amplification of gene loss in domesticated common bean could reflect a classic bottleneck effect[18] rather than natural selection[19]. This suggestion is supported by the fact that the M1/Jalisco-Durango and A2/Peru races have more PAVs than the other domesticated subgroups in their respective gene pools (Fig. 2d), and this difference is especially noticeable among the Andean subgroups. This was corroborated by nucleotide diversity analysis applied to the 1,451,663 core SNPs (Supplementary Fig. 5 and Supplementary Data 5), and agrees with a recent hypothesis proposing that the M1/Durango-Jalisco and A2/Peru races were the first domesticated Mesoamerican and Andean populations from which the M2, A1 and A3 races arose during a secondary domestication phase[13].

To study the differentiation between gene pools, we analyzed the PAV matrix for American domesticated accessions by using Fisher's exact test to compare the Mesoamerican and Andean populations. We found that more than 60% of the PAVs (5223) differ significantly in terms of frequency ($p < 0.05$) between the two gene pools. These included 721 diagnostic PAVs, indicating that they are present in one population with a frequency of one and completely absent in the other (frequency of zero). In detail, 90% (650) of the diagnostic PAVs were fixed in the Mesoamerican gene pool and the remaining 10% (71) in the Andean gene pool (Supplementary Data 6). GO enrichment analysis applied to the 721 diagnostic genes revealed enrichment in processes related to metabolism (GO:0008152), detoxification (GO:0098754), and responses to stimuli (GO:0050896) (Supplementary Fig. 6). Interestingly, none of these PAVs were found to be diagnostic between gene pools in Europe (Supplementary Data 6), and when a PAV-based Fisher's exact test was applied to the subset of 114 European accessions, we did not detect any diagnostic genes between the Mesoamerican and Andean gene pools (Supplementary Data 7). These outcomes reflect the extensive inter-gene-pool hybridization in European germplasm and confirm its key role in the adaptation of common bean to new agricultural environments[13,20].

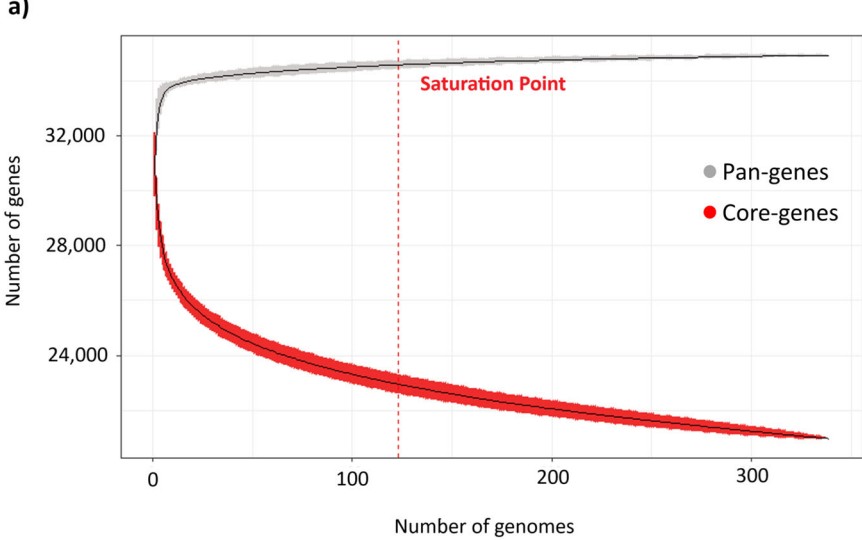

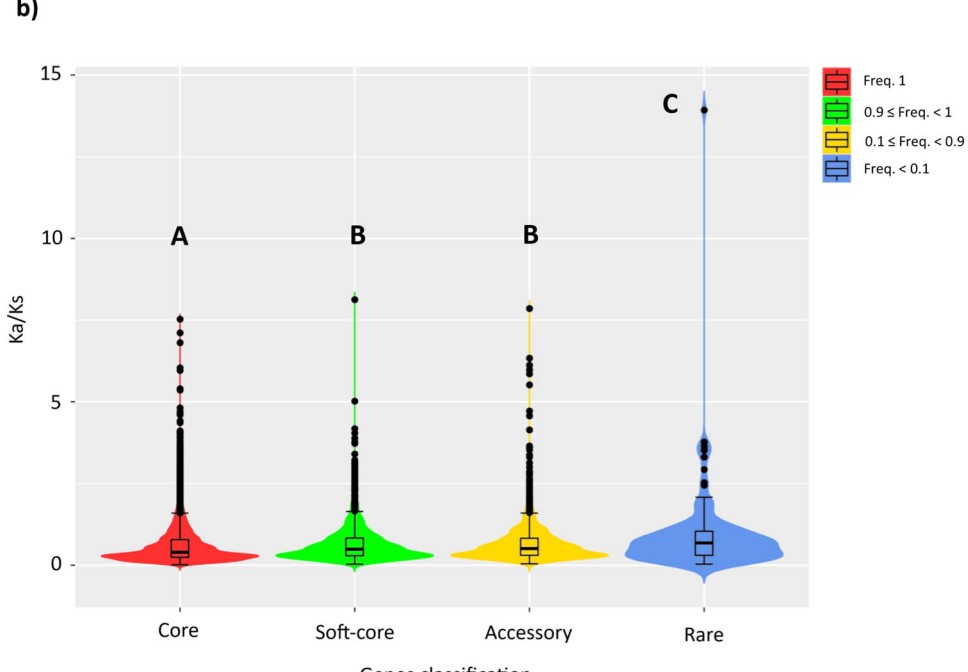

**Fig. 1 | Characterization of the common bean pan-genome. a** Pan-gene and core gene size calculation. The growth curve of pan-genes (gray) reached saturation point (99%, 33,997 genes) when 125 individuals were included, as indicated by the dashed red line. In contrast, the growth curve of core genes (red) diminished with the addition of each genotype. Data for pan-genes and core genes are presented as mean values ± SD. **b** Violin plots showing analysis of variance (ANOVA) related to the ratio of non-synonymous to synonymous mutations (Ka/Ks) in core genes and PAVs categorized by frequency (soft-core, accessory, and rare). Box plots represent minimum, first quartile, median, third quartile, and maximum. Sample sizes (*n*) for each category are as follows: core *n* = 16,264, soft-core *n* = 2672, accessory *n* = 2156, and rare *n* = 140. Violin plots display the data distribution for each gene category, with significant differences indicated by different letters above the plots, based on a Tukey–Kramer HSD post hoc test. Additionally, statistical significance was determined by applying a two-sided pairwise Wilcoxon test. Detailed statistics are available in Supplementary Table 3. Source data are provided as a Source Data file.

To investigate the influence of PAVs on important trait (flowering time) variations and identify candidate genes associated with them, we conducted a PAV-based genome-wide association study (GWAS) involving 218 American and European domesticated genotypes. Using previously reported phenotypic data[13], we identified 39 significant association events ($p \le 7.07E-06$) correlated with day-to-flowering and photoperiod sensitivity. These associations were linked to 35 candidate PAVs, highlighting their probable involvement in the regulation of floral transition (Supplementary Data 8), one of the major diversification traits that defines the adaptation of plant populations to different agro-ecological conditions. An interesting example is the GWAS peak

associated with flowering time and photoperiod sensitivity located on Phvul.003G185200 (Chr03:40,838,810-40,850,729) (Fig. 3a). This PAV is orthologous to the *HDA5* gene in *Arabidopsis thaliana*, which encodes a deacetylase. Notably, *A. thaliana* mutants with impaired *HDA5* expression patterns display late-flowering phenotypes due to the upregulation of two floral repressor genes, namely *FLOWERING LOCUS C* (*FLC*) and *MADS AFFECTING FLOWERING 1* (*MAF1*)[21]. It is noteworthy that common bean genotypes lacking PAV Phvul.003G185200 exhibit early-flowering phenotypes compared to accessions carrying this gene (Fig. 3b). Additionally, the presence of Phvul.003G185200 in all Mesoamerican accessions contrasts with its limited presence (only 18%)

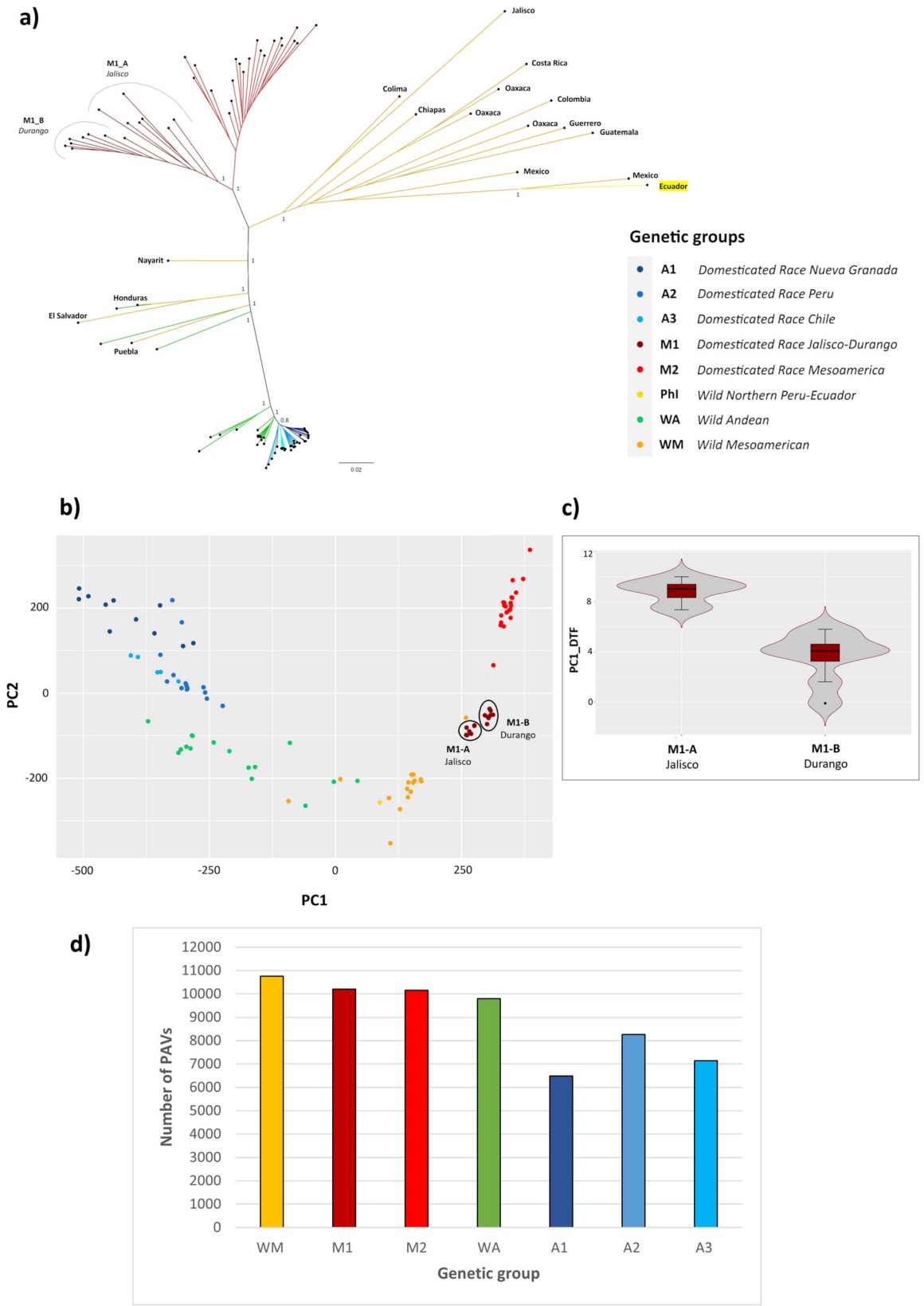

in the Andean gene pool (Fig. 3c). The divergent distribution of Phvul.003G185200 in the Mesoamerican and Andean gene pools may suggest an adaptive response associated with its loss during population differentiation. Furthermore, we found that nine of the 35 candidate PAVs from the GWAS display signatures of selection in various comparisons: specifically, two PAVs differing between wild and domesticated Mesoamerican populations and seven PAVs differing between wild and domesticated Andean populations. Overall, although the majority (59%) of the candidate PAVs were located on the reference genome, 41% were situated on the NRRs (Supplementary Data 8), reaffirming the ability of the pan-genome to identify functional variants associated with economically and evolutionarily important traits.

**Fig. 2 | Population structure of *P. vulgaris*. a** Neighbor-joining (NJ) phylogenetic tree constructed using only SNPs located in core genes (bootstrap = 1000). **b** PAV-based principal component analysis (PCA). **c** Violin plots showing the analysis of variance (ANOVA) for the first principal component representing days to flowering and photoperiod sensitivity (PC1_DTF) in the M1/Jalisco-Durango races by splitting the accessions into two clusters based on PCA and the NJ tree. The PC1_DTF trait was derived from the multivariate PCA analysis of days to flowering and photoperiod sensitivity data collected in 10 different environments[13]. Box plots represent minimum, first quartile, median, third quartile, and maximum. Sample sizes ($n$) for each category are as follows: Jalisco $n = 7$, Durango $n = 8$. Statistical significance was determined by applying a two-sided Student's $t$ test. Detailed statistics are available in Supplementary Table 4. **d** Bar chart showing the number of PAVs per genetic group, representing the number of genes present across the sampled genotypes. Source data are provided as a Source Data file.

## Pan-genome shrinkage during wild expansion to South America

One of the most striking outcomes we observed was the difference in pan-genome size between the Mesoamerican and Andean gene pools (Fig. 4a). We calculated the total number of PAVs per individual and found that accessions from the same gene pool clustered together in separate groups, with Mesoamerican accessions exhibiting a higher number of PAVs per individual (i.e., a greater number of genes present) compared to those from the Andean gene pool (Fig. 4b, c and Supplementary Table 6). This reduction in pan-genome size may reflect genetic drift and the two sequential bottlenecks that occurred solely in the Andean population[12]. To better understand the roles of different evolutionary forces in shaping the PAV content of the Mesoamerican and Andean gene pools, and to distinguish between the effects of adaptation, population demography and history, we initially considered a panel of wild genotypes representing the entire geographical distribution in Latin America. We applied bivariate fit analysis and found a significant correlation ($p < 0.0001$) between the number of PAVs per individual and the latitude. Analysis of variance, in which wild individuals were grouped by latitude followed by spatial interpolation, revealed the progressive loss of genes ranging from the accessions of Northern Mexico to those of Northwestern Argentina (Fig. 5a, b and Supplementary Table 7). Furthermore, FST analysis of PAVs comparing Mesoamerican and Andean wild populations may suggest selection for gene loss during wild range expansion (Fig. 5c and Supplementary Data 9). We found that 64% of the PAVs in the top 5% of the FST distribution (FST ≥ 0.85; candidate PAVs) are missing from the wild Andean gene pool. This high rate of absences exceeds that observed in the entire variable genome (25%), demonstrating a more than twofold increase. This difference was statistically validated using bootstrap resampling, strongly suggesting that gene loss during the process of wild differentiation was not a random occurrence but the evident outcome of selective forces (Supplementary Figs. 7 and 8). Moreover, functional annotation of the candidate PAVs revealed the enrichment of genes involved in pollen germination, innate immunity, abiotic stress tolerance, and root hair growth, indicating a potential adaptive role during wild range expansion (Supplementary Data 10). Overall, our findings suggest that selective pressure favoring the loss of genes involved in adaptive mechanisms, coupled with the influence of genetic drift resulting from the founder effect, may have contributed to the shrinking of the Andean pan-genome during wild differentiation.

## Footprints of selection for gene loss during domestication

The PAVs putatively shaped by selection during domestication in Mesoamerica and the Andes revealed further evidence that gene loss underpinned the successful adaptation of the American common bean. FST analysis was applied to PAVs in wild and domesticated forms (separately for each gene pool) with only PAVs in the top 5% of the FST distribution considered as candidates (Supplementary Data 11 and 12). We found 610 PAVs potentially under selection in the Mesoamerican population (FST ≥ 0.30) and 497 in the Andean population (FST ≥ 0.27). Moreover, functional annotation of the candidate PAVs revealed the enrichment of genes associated with domestication syndrome and adaptive traits such as dormancy, floral transition, light acclimation, defense, and symbiotic interactions (Supplementary Data 13 and 14). Importantly, the candidate Phvul.003G265200 (Chr03: 50,365,995-50,368,501) is orthologous to 11 members of the plant Rho GTPase subfamily (ROP), including *ROP6* encoding a small Rho-like GTP binding protein. This GTPase subfamily is required for symbiotic interactions[22–24], and in the plasma membrane of *Lotus japonicus* cells it interacts directly with NOD FACTOR RECEPTOR 5, one of two nodulation factor receptors essential for nodule formation during symbiosis[25]. From our analysis, Phvul.003G265200 is a putative selected PAV (FST = 0.50) for the Mesoamerican gene pool, whose presence declined by more than 60% during progression from the wild (0.94) to the domesticated (0.25) population (Supplementary Data 11). Specificity is one possible explanation for the biological importance of the loss of Phvul.003G265200 in Mesoamerican domesticated genotypes. In common bean populations, different genotypes preferentially associate with specific strains of nitrogen-fixing bacteria. Consequently, the absence of Phvul.003G265200 in domesticated genotypes may increase the flexibility of symbiotic interactions, enabling adaptation to diverse environmental conditions and facilitating interactions with a broader range of symbiont partners. This hypothesis parallels the cost–benefit trade-off commonly observed among resistance genes. Similar to resistance genes, the absence of Phvul.003G265200 may confer advantages by mitigating potential fitness costs associated with specific symbiotic interactions. By losing specificity and expanding the spectrum of symbiotic partners, common bean populations lacking this gene may achieve greater adaptability and resilience in fluctuating environments. As for Phvul.003G265200, 72% of PAVs putatively under selection (437 genes) in the Mesoamerican population (Fig. 6a) and 80% (398 genes) in the Andean one (Fig. 6b), were present with a lower frequency in domesticated than wild populations. When considering all PAVs, the percentage of genes present at lower frequencies in domesticated populations fell significantly to 28% ($p < 2.2 \times 10^{-16}$) for the Mesoamerican gene pool and 43% ($p < 2.2 \times 10^{-16}$) for the Andean one (Fig. 6a, b). On the other hand, we observed no significant differences in absences between the wild and domesticated populations for both gene pools (Fig. 6a, b). Overall, these findings suggest that selection during domestication led to a reduction in gene presence. But unlike the range expansion of wild populations, where we found footprints of selection for absences, we did not find any evidence of complete gene loss due to selection during domestication. This may reflect the different evolutionary timescales involved: wild differentiation occurred ~150,000 years ago whereas domestication was much more recent at ~8000 years ago. These findings are consistent with previous observations that selection during the domestication of common bean in Mesoamerica has directly affected the transcriptome, leading to a ~20% decrease in gene expression levels attributed to loss-of-function mutations[19]. We also detected 29 PAVs with high FST values in common between the Mesoamerican and Andean gene pools, and these are mainly associated with the tryptophan metabolic pathway. Tryptophan is a precursor of key secondary metabolites such as auxin, serotonin, and melatonin. These compounds play diverse roles in plant physiology, influencing processes such as seed germination, root development, senescence, and flowering. Additionally, they contribute to biotic and abiotic stress responses[26]. We found that ~86% of these PAVs in both gene pools declined in terms of presence during the progression from wild to domesticated accessions (Supplementary Table 8). This may indicate a pattern of genomic convergence for the loss of key adaptive genes between the Mesoamerican and Andean populations during their parallel domestication events.

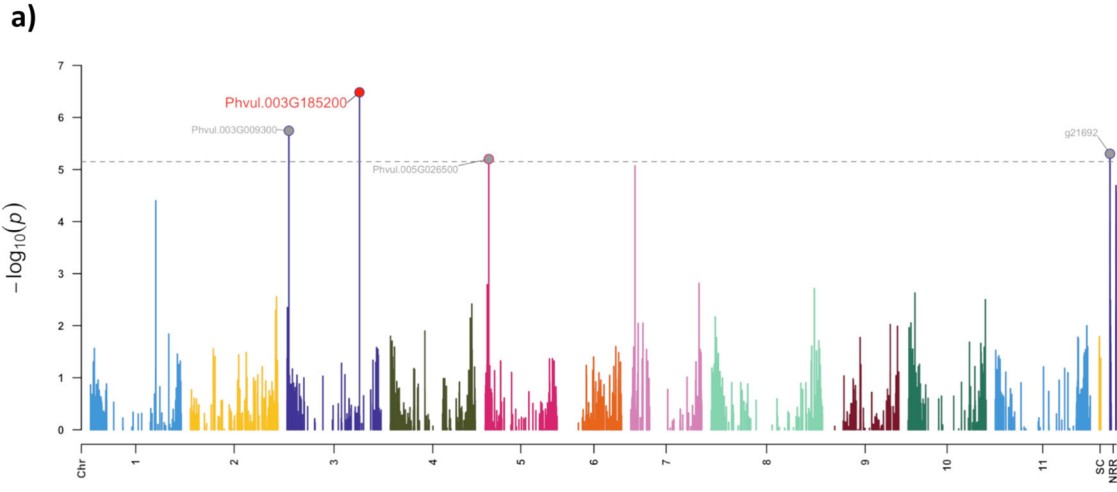

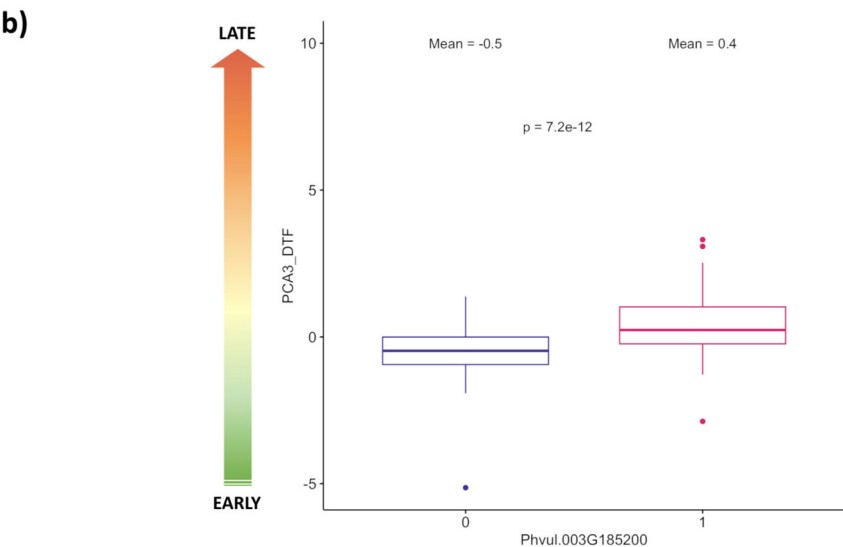

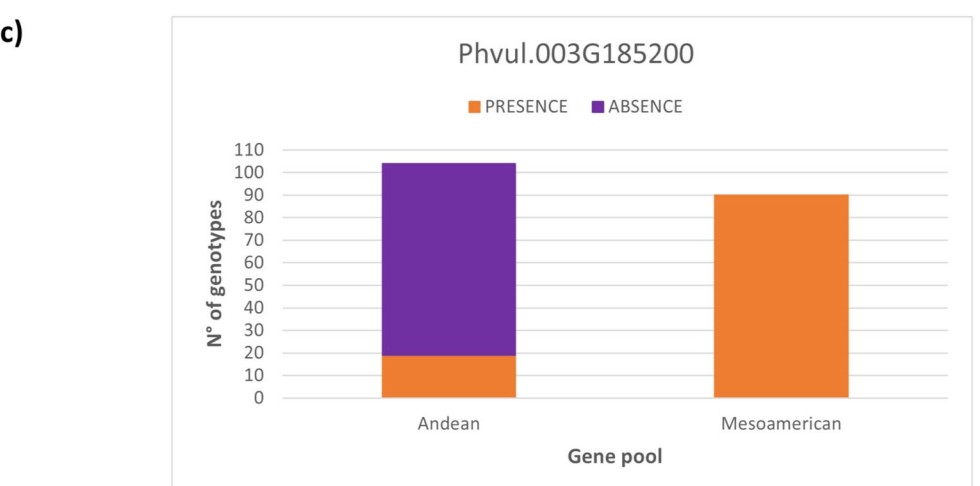

## Implications for legume research and breeding

The genotypes selected for this study encompass wild and domesticated forms, ensuring that the pan-genome comprehensively captures the extensive genetic variation within this species. PAV analysis provided insight into the evolutionary dynamics of pan-genome adaptation, including signatures of selection for complete gene loss during wild differentiation between the Mesoamerican and Andean gene pools, contributing to the smaller pan-genome of the Andean population. We also identified selection footprints for gene loss during Mesoamerican and Andean domestication, causing reductions in gene presence in domesticated populations compared to their wild counterparts. Interestingly, candidate genes that have been entirely or

**Fig. 3 | Case study of Phvul.003G185200. a** PAV-based GWAS for flowering time in American and European domesticated genotypes. The complex PCA3_DTF was derived from the multivariate PCA analysis of days to flowering and photoperiod sensitivity data collected in 10 different environments[13]. The most significant PAV-trait association, located on chromosome 3 (Phvul.003G185200), was accompanied by three minor associations, spanning chromosomes 3 and 5, as well as non-reference regions (NRRs). **b** Boxplots of the trait "PCA3_DTF" by PAVs (1 = presence; 0 = absence) at locus Phvul.003G185200. Box plots represent minimum, first

quartile, median, third quartile, and maximum. Higher values of PCA3_DTF indicate late flowering phenotypes whereas lower values indicate early flowering phenotypes. The significant difference between groups carrying the 0 allele ($n = 85$) and the 1 allele ($n = 114$) was tested by applying a two-sided Wilcoxon test. **c** Bar chart showing the proportions of presence/absence for Phvul.003G185200 in the Mesoamerican and Andean gene pools. Source data are provided as a Source Data file.

partially lost appear to be involved in important adaptive mechanisms, such as flowering time, symbiosis, biotic and abiotic stress tolerance, and root hair growth. Gene loss is considered functionally equivalent to other loss-of-function mutations, such as premature stop codons, providing an important and abundant source of adaptive phenotypic diversity[19,27–32]. Moreover, variations in genome size have been described between different populations of microorganisms and plants[33–35]. For example, in contrast to their European native counterparts, invasive plants have smaller genomes resulting in phenotypic effects that could enhance their invasive potential[36]. Similarly, genome size variations within the *Zea mays* species during the post-domestication process revealed that maize landraces have significantly smaller genomes than their closest wild relatives, the teosintes[37]. However, it is still unclear to what extent genome size variation is shaped by natural selection. Here, our results suggest that under the influence of specific and diverse agro-ecological pressures, the relinquishment of particular genes can confer a selective advantage. This may be relevant when populations face selective pressure from radical environmental changes, such as the expansion of wild common bean from Central Mexico's warm and humid climate to higher and cooler altitudes in the Andes. Our research establishes a paradigm in which natural selection drives gene loss, favoring adaptation over stochastic responses. Mutations are more likely to cause a loss rather than a gain of function, so adaptive gene loss provides a rapid evolutionary response to environmental changes. This could have profound implications for our understanding of crop adaptation in response to climate change. The common bean pan-genome is a valuable starting point that will lead to a deeper understanding of the genetic variations and genome dynamics responsible for key adaptive traits in food legumes, and will accelerate breeding programs to improve food legume crops.

## Methods

### Sources of genetic diversity
The pan-genome was constructed from five high-quality genomes representing wild and domesticated forms belonging to the Mesoamerican and Andean gene pools. The *P. vulgaris* reference genome G19833 v2.1 was downloaded from Phytozome[38], the genomes of BAT93 and JaloEPP558 were provided by the INRAE Institute, and the genomes of MIDAS and G12873 were sequenced and assembled de novo specifically for this study (Supplementary Table 9). We also integrated 339 representative low-coverage WGS common bean genotypes, including 220 domesticated and 10 wild accessions from previous studies[11,13]. The remaining 109 accessions were multiplied in the greenhouse, and DNA extracted from young leaves was used for sequencing (Supplementary Data 15). See "Data availability" statement.

### Plant growth and DNA extraction
MIDAS and G12873 single seed descent (SSD) genotypes were multiplied in the greenhouse. For both samples, high-molecular-weight (HMW) DNA was extracted from 2 g of young leaves following the method described in ref. 39. Briefly, tissue grounded in liquid nitrogen was resuspended in MEB extraction buffer (1 M 2-methyl-2,4-pentanediol (MPD), 10 mM PIPES-KOH, 10 mM MgCl2, 2% polyvinylpyrrolidone (PVP10), 10 mM sodium metabisulfite, 5 mM β-mercaptoethanol, 0.5% sodium diethyldithiocarbamate, 6 mM EGTA, 200 mM L-lysine-HCl, pH

5.0) and filtered through 100 μm and 40 μm cell strainers. After the addition of Triton X-100 (0.5%), the homogenate was incubated 30' on ice and then centrifuged at $800 \times g$ for 20' at 4 °C. Nuclei were washed four times in MPDB buffer (0.5 M 2-methyl-2,4-pentanediol, 10 mM PIPES-KOH, 10 mM MgCl$_2$, 0.5% Triton X-100, 10 mM Sodium metabisulfite, 5 mM β-mercaptoethanol, pH 7.0) and purified through a gradient of 37.5% Percoll (centrifugation at $650 \times g$ for 1 h). Purified nuclei were washed twice in MPDB buffer, collected by centrifugation at $2500 \times g$ for 10' at 4 °C, and finally resuspended in TE buffer (10 mM Tris-HCl, 1 mM EDTA, pH 8). DNA was extracted from the isolated nuclei pellets using the Qiagen Genomic tip-100 (Qiagen) following the manufacturer's instructions. DNA quality was evaluated according to Oxford Nanopore Technologies (ONT) requirements. Specifically, purity was assessed using a NanoDrop 1000 spectrophotometer (Thermo Fisher Scientific), the concentration was determined using a dsDNA Broad Range Assay Kit with Qubit 4.0 (Thermo Fisher Scientific), and the fragment size (≤400 kb) was determined using the CHEF Mapper electrophoresis system (Bio-Rad Laboratories). Fragments <25 kb were removed using the Short Reads Eliminator kit (Circulomics) leaving 75% of the DNA from the MIDAS samples and 95% from the G12873 samples. *P. vulgaris* genotypes of BAT93 and JaloEEP558 were sowed in soil and grown in a growth chamber at 23 °C and 75% humidity with a 16-h photoperiod under fluorescent tubes (166lE). Young trifoliate leaves of BAT93 and JaloEEP558 genotypes were collected and flash-frozen in liquid nitrogen. Three days before sampling, plants were dark-treated to optimize the extraction of HMW DNA. The 109 SSD accessions were multiplied in the greenhouse and young leaves were collected in silica gel for drying and subsequent DNA extraction using the DNeasy 96 Plant kit (Qiagen) according to the manufacturer's instructions. For each sample, 50–70 mg of dried leaf material was pulverized with a Tissue-Lyser II (Qiagen) at 30 Hz for 6 min. The DNA quality and quantity were evaluated using a Nano-Photometer NP80 (Implen), and the concentration was determined using a Qubit BR dsDNA assay kit (Thermo Fisher Scientific).

### Sequencing of low-coverage WGS accessions
DNA libraries for all samples were prepared using a KAPA Hyper Prep kit and PCR-free protocol (Roche). For each genotype, 200 ng of DNA was fragmented by sonication using a Covaris S220 device (Covaris). WGS DNA libraries were generated using a 0.7–0.8× ratio of AMPure XP beads for final size selection. Libraries were quantified using the Qubit BR dsDNA assay kit, and equimolar pools were quantified by real-time PCR against a standard curve using the KAPA Library Quantification Kit (Kapa Biosystems). Libraries were sequenced on the Nova-Seq 6000 Illumina platform, producing 15–30 million 150-bp paired-end reads per sample.

### Sequencing and assembly of the MIDAS and G12873 genomes
Following quality control and priming according to ONT specifications, libraries were sequenced on a MinION device with a SpotON flow cell (FLO-MIN106 R9.4.1-Rev D). Two libraries were prepared for each genotype according to the SQK-LSK109 ligation sequencing protocol (ONT) with minor adjustments. Each library was loaded twice, and the flow cell was washed using the Flow Cell Wash Kit (ONT). Illumina PCR-free libraries were prepared starting with 1 μg of fragmented genomic DNA using the KAPA Hyper prep protocol. This process involved

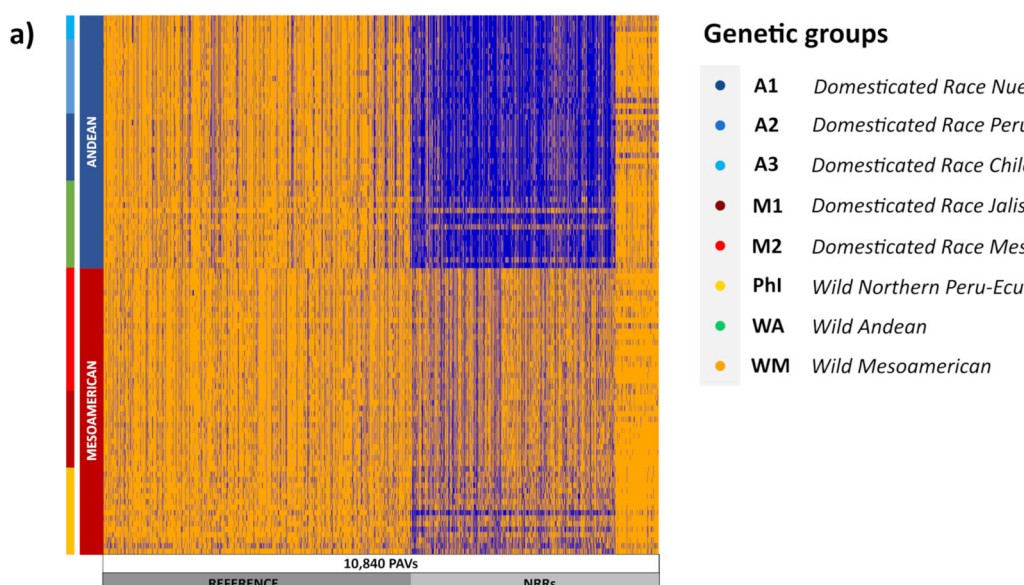

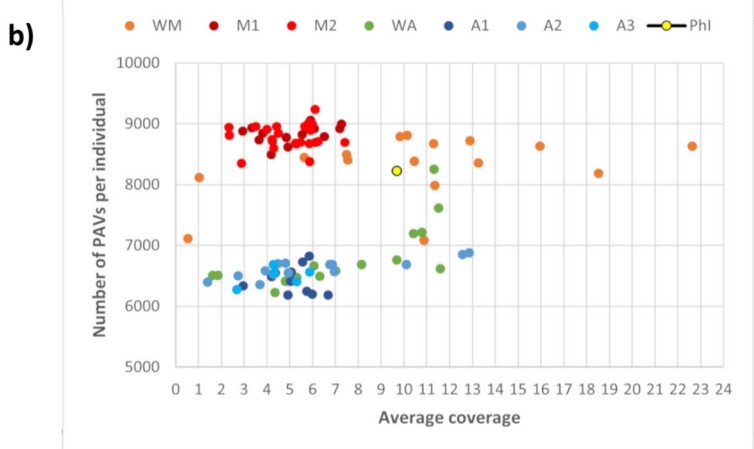

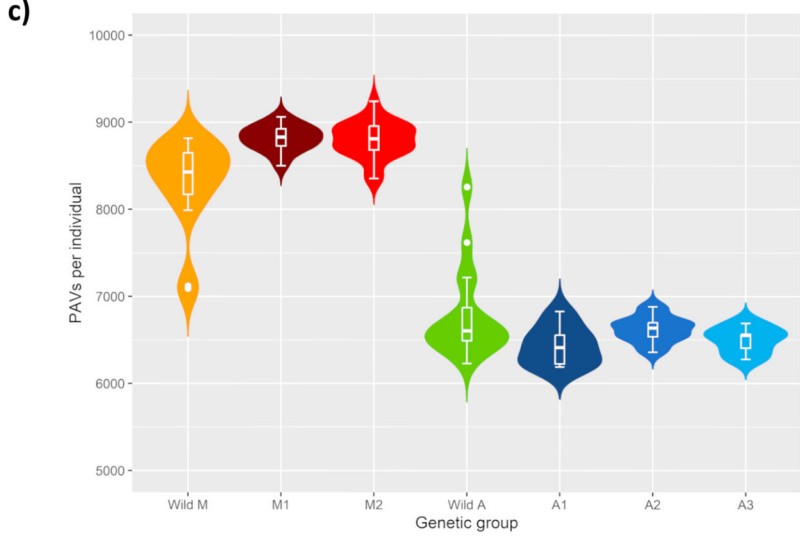

extending the adapter ligation time to 30 min and conducting post-clean-up size selection using 0.7× AMPure XP beads. Library concentration and size distribution were assessed using a Bioanalyzer 2100 with high-sensitivity DNA reagents and chips. Sequencing was performed on a NovaSeq6000 instrument to generate 150-bp paired-end reads. MIDAS and G12873 whole-genome assemblies were generated by nanopore sequencing based on 26 Gb (50-fold coverage) and 36 Gb (69-fold coverage), respectively. Raw nanopore reads were corrected using Canu v2.1[40] and the resulting corrected reads were assembled de novo using wtdbg2 v2.5[41]. Draft assemblies were refined by iterative polishing using long reads (Racon v1.4.3 and Medaka v1.0.3)[42] and short reads (three rounds of Pilon v1.23)[43]. Completeness

**Fig. 4 | Evolution of the common bean pan-genome. a** Heat map illustrating the distribution of 10,840 PAVs across the final pan-genome, distinguishing between those mapped on the reference genome *Phaseolus vulgaris* v2.1 G19833 and those located on non-reference regions (NRRs). The distribution is displayed in relation to the common bean subgroups. Orange indicates gene presence and blue indicates gene absence. **b** Scatterplot showing the number of PAVs per individual (*y*-axis), representing the number of genes present across the sampled genotypes, in relation to the coverage (*x*-axis) of each genotype. **c** Violin plots representing the analysis of variance (ANOVA) for the number of PAVs per individual by genetic group. Sample sizes (*n*) for each group are as follows: Wild M $n = 16$, M1 $n = 15$, M2 $n = 21$, Wild A $n = 16$, A1 $n = 11$, A2 $n = 14$, and A3 = 5. Box plots represent minimum, first quartile, median, third quartile, and maximum. Statistical significance was determined by applying a two-sided Wilcoxon test. Supplementary Table 6 contains detailed statistics. Source data are provided as a Source Data file.

was assessed using BUSCO v4.1.2[44] and the Fabales_odb10 dataset (Supplementary Table 10).

## Sequencing and assembly of the BAT93 and JaloEEP558 genomes

HMW DNA from genotypes BAT93 and JaloEEP558 was sequenced using the PacBio Sequel II system by GENTYANE (INRAE Clermont-Ferrand, France). A total of 21.09 and 29.35 Gb of PacBio HiFi reads was generated from BAT93 and JaloEEPP558, respectively. The PacBio HiFi reads were assembled de novo into contigs using HiFiasm v0.9.0 with default parameters[45].

## Orthologous/paralogous analysis and clustering threshold settings

To incorporate the Andean and the Mesoamerican gene pools into the pan-genome, precise differentiation between orthologous and paralogous genes required a meticulous strategy to preserve solitary orthologs and all paralogous counterparts. The relationship between orthologous genes was calculated using minimap2 v2.17[46] to align the MIDAS and G12873 genome assemblies using the open reading frames (ORFs) of 2,330 complete single-copy Benchmarking Universal Single-Copy Orthologs (BUSCO) genes in common between the reference genome G19833 v2.1, MIDAS, and G12873 (Supplementary Data 16). The percentage identity was calculated for each ORF based on the number of matches in the alignments as a proportion of ORF length. The relationship between paralogous genes was calculated using the three most abundant gene families (OG0000273, OG0000328 and OG0000085) in the *P. vulgaris* G19833 v2.1 reference genome, composed of 26, 37 and 42 genes, respectively. An all-versus-all comparison between the members of the same family was implemented using minimap2. The percentage identity was calculated for each gene family by dividing the number of matches in the alignments by the reference gene ORF length and then averaging the identity percentages for each family. Finally, the results of both tests were used to establish a clustering threshold of 90% to retain only one orthologous and all paralogous genes in the pan-genome (Supplementary Data 17).

## Pan-genome construction

We used a paired genome alignment strategy[47] involving a non-iterative approach (independent alignment of the reference genome to the other high-quality genomes). This ensured the preservation of information regarding the origins of the NRRs. Specifically, the G19833 v2.1 reference genome was independently mapped onto the four high-quality genomes (MIDAS, G12873, BAT93 and JaloEPP558) with minimap2 v2.17 using the alignment pre-set -x asm5, which considers regions with an average divergence <5%. Subsequently, the resulting bam files from the four alignments were converted to delta files, and structural variants were identified using Assemblytics v1.2.1[48]. Among these variants, only deletions were selected as NRRs[47]. Additionally, we used samtools depth v1.1[49] on the bam files to identify uncovered contigs unique to the four high-quality genomes, which were then extracted and also classified as NRRs. Then, deletions and uncovered contigs were independently filtered for a minimum length of 1 kb and clustered using CD-HIT-EST v4.8.1[50] with a sequence identity of 90% *(-c 90)*, as described above for the orthologous and paralogous genes. To validate the accuracy of the detected NRRs and ensure they reflect gene content rather than allelic variation, we conducted a comparative analysis using highly conserved BUSCO genes. In detail, we examined the entire set of 4,947 MIDAS and 4,812 G12873 BUSCO genes, shared with the reference genome G19833 v2.1, within the NRRs using BLASTp. The outcome revealed that few genes (seven in MIDAS and 37 in G12873) were identifiable within the NRRs, confirming the reliability and accuracy of our NRRs detection method. The NRRs were incorporated into the *P. vulgaris* G19833 v2.1 reference genome to provide a preliminary pan-genome. Subsequently, Illumina data representing the 339 low-coverage WGS common bean accessions were trimmed with fastp v0.21.0[51] and aligned to the preliminary pan-genome using bowtie2 v2.3.5.1[52] with default parameters. The unmapped reads were extracted using samtools v1.11, pooled, assembled using MaSuRCA v3.4.2[53] with default parameters, and added to the preliminary pan-genome. The integration of the reference genome with the NRRs from the four high-quality genomes, in conjunction with the NRRs derived from the 339 WGS genotypes, led to the development of the final common bean pan-genome. To exclude putative contaminants and/or organelle sequences, NRRs were compared to the NCBI non-redundant nucleotide database using BLASTn, considering a minimum of 80% identity and 25% coverage, leading to the removal of 1194 sequences. Overall, we identified 61,680 added sequences, 88% of which reflected the mapping of the 339 low-coverage WGS accessions. The remaining 12% were identified by comparing the reference genome G19833 v2.1 independently with the other four high-quality genomes (Supplementary Table 1).

## RNA sequencing

RNA sequencing (RNA-Seq) was conducted on leaf tissues obtained from genotypes G12873 and MIDAS cultivated under controlled greenhouse conditions (relative humidity ~70% and an average night/day temperature of 25 °C). Leaf samples were collected at two stages of pod development, specifically at 5 and 10 days. RNA was extracted from frozen tissues[19] and non-directional Illumina RNA-Seq libraries were prepared and sequenced using the Illumina HiSeq 2500 platform, generating 125-bp paired-end reads.

## Pan-genome annotation

Repetitive sequences were identified and soft-masked using Repeat-Modeler v2.0.2[54] and RepeatMasker v4.1.2-p1[55], respectively. For pan-genome annotation, we adopted a hybrid approach. This involved the ab initio prediction of protein coding genes with Augustus v3.3.3[56], complemented by extrinsic supporting evidence in the form of *P. vulgaris* RNA-Seq data from this study and elsewhere[19] as well as protein sequences from *P. vulgaris* and closely related species such as *Medicago truncatula* and *Glycine soja*. The protein sequences and RNA-Seq data were aligned to the pan-genome with Hisat2 v2.2.1[57] and Genome Threader v1.7.1[58], respectively. BUSCO genes in the Fabales_odb10 database were used to train the model for the Augustus predictor. The predicted genes were then scanned with InterProScan v5.46-81.0[59] for the presence of protein domains. The InterProScan results were filtered to remove genes with transposon-related domains, ensuring that only those with at least one recognized protein domain were retained in the annotation. The filtered proteins were compared to the pan-genome with BLASTp v2.12.0[60] and filtered by the best hits. The predicted genes were clustered with the proteins of all species considered in the annotation using OrthoFinder v2.5.4[61]. Finally, functional annotation was achieved by integrating information

a)

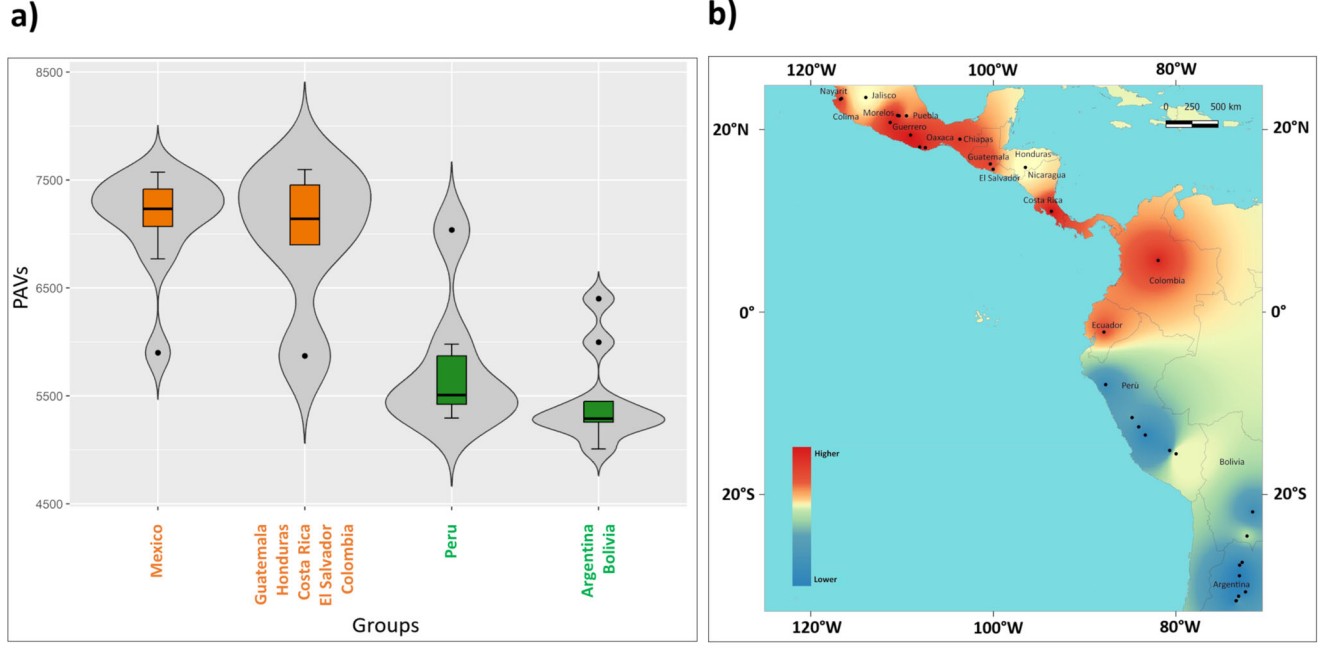

c)

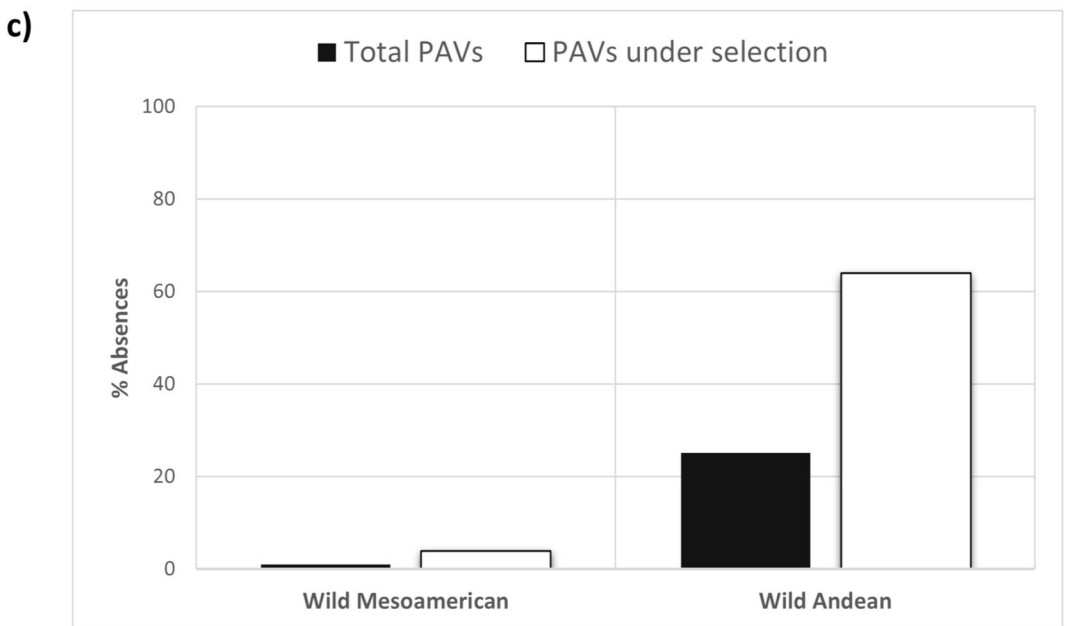

**Fig. 5 | Selection for adaptive gene loss during the expansion of wild common bean. a** Violin plots showing the analysis of variance (ANOVA) for the number of PAVs per individual based on grouping the wild Mesoamerican and Andean accessions according to latitude coordinates. Wild Mesoamerican genotypes are colored orange, while wild Andean genotypes are green. Sample sizes (*n*) for each category are as follows: Mexico *n* = 11, Guatemala, Honduras, Costa Rica, El Salvador, and Colombia *n* = 5, Peru *n* = 6, Bolivia and Argentina *n* = 9. Box plots represent minimum, first quartile, median, third quartile, and maximum. Statistical significance was determined by applying a two-sided Tukey–Kramer HSD post hoc test. Detailed statistics are available in Supplementary Table 7. **b** Spatial interpolation of wild common bean genotypes based on the number of PAVs per individual. Dark red regions indicate a higher number of PAVs and blue regions a lower number of PAVs. Latitude and longitude values are indicated in degrees using the geographic coordinate system. **c** Bar charts showing the proportions of absences found for the subset of PAVs putatively under selection during the wild expansion (white) and for the entire variable genome (black). Source data are provided as a Source Data file.

about orthologous genes and identifying functional domains using a custom script.

## PAV calling
We developed a specific threshold for PAV calling, termed the MIN threshold, as an alternative to the commonly used 0.05 threshold based on gene coverage[62,63]. The MIN threshold is based on the minimum coverage value of 1000 randomly selected BUSCO genes (ORFs) for each accession, allowing for the definition of an accession-specific threshold for calling genes as present. Specifically, Illumina data representing the 339 low-coverage WGS accessions were aligned to the pan-genome using bowtie2 v2.3.5.1 and the coverage of 1000

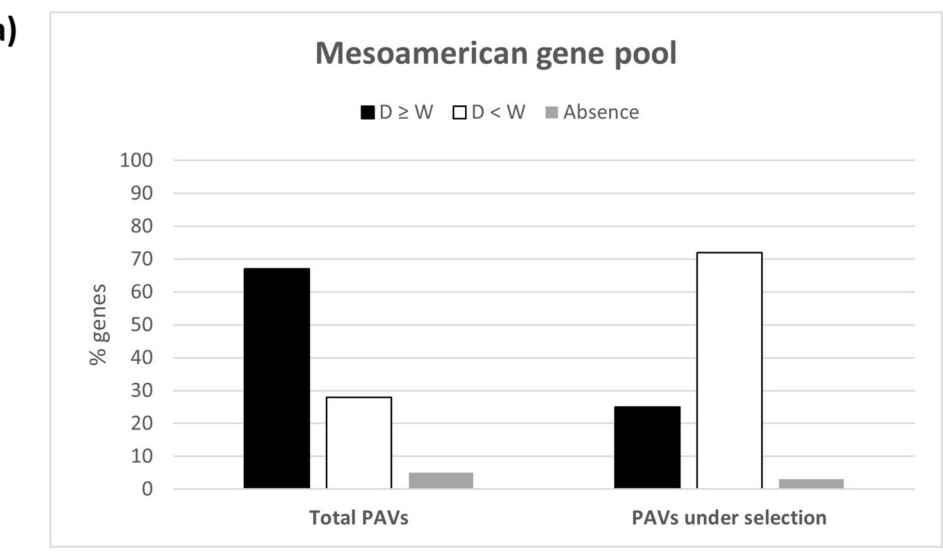

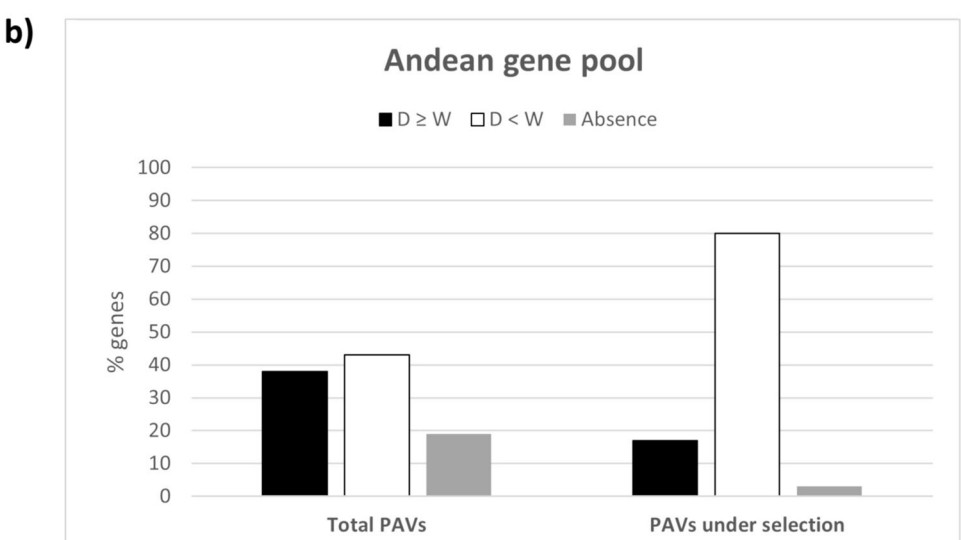

**Fig. 6 | Adaptive reduction effects during the domestication of the common bean. a** Bar chart showing the proportions of presence/absence in the Mesoamerican gene pool for the entire variable genome (left) and for the subset of PAVs putatively under selection (right) between wild and domesticated populations (right). **b** Bar chart showing the proportions of presence/absence in the Andean gene pool for the entire variable genome (left) and for the subset of PAVs putatively under selection between wild and domesticated populations (right). In both charts, the presence values are divided based on frequency (≥/<) in the comparison between wild and domesticated populations. Source data are provided as a Source Data file.

randomly selected BUSCO genes (ORFs) was calculated for each accession using samtools v1.11 (Supplementary Data 18). PAV calling thresholds were defined for each accession according to the minimum coverage value of the 1000 BUSCO genes. To avoid bias caused by a few underrepresented BUSCO genes, the 10 least-covered genes in each accession were discarded. The identified genes were classified based on their frequency as core genes if present in all the accessions or PAVs if partially shared or private to a single genotype (Supplementary Table 2 and Supplementary Data 1).

### Pan-genes and core genes size calculation
The curves describing the pan-genome and core genome sizes were evaluated by considering 1000 random orders of the 339 genotypes with a custom script. The orders were chosen randomly among all possible permutations (n!, where $n = [1339]$). For each ordering, the gene sets of the accessions were progressively added to the total genome size without considering the genes already present in the total set. The same procedure was applied for the core genome size, but the gene sets were intersected when each genome was added, thus

keeping only the genes in common for each iteration (Supplementary Data 19 and 20).

### Variant calling
SNVs and InDels were called with bcftools v1.10.2[64] based on the alignment of 339 accessions with the pan-genome using bowtie2 v2.3.5.1. We used bcftools mpileup v1.10.2 to generate a genotype likelihood table. Variants were identified using bcftools call v1.10.2 and the pileup table, producing the raw VCF file. During the pileup step, the filtering parameter for minimum mapping quality (-q) was set to 20[47].

### Data analysis
Pan-genome analysis focused on a representative subset of 99 well-characterized accessions among the original 339, including wild and American domesticated forms. In some cases, we also analyzed the subset of 114 European domesticated accessions (Supplementary Data 15).

For GO enrichment, the annotated core genes and PAVs in the pan-genome were analyzed using the buildGOmap R function to infer

indirect annotations and generate data suitable for clusterProfiler[65,66]. Diagnostic genes were analyzed using Metascape[67]. *A. thaliana* orthologs were identified using OrthoFinder[61] and by comparing all protein sequences from *P. vulgaris* (v2.1) and *A. thaliana* (TAIR10). For PCA, the PAV matrix (1/0) was analyzed using the logisticPCA package in R[68].

ANOVA within subgroup M1 was carried out using the first principal component related to days-to-flowering and photoperiod sensitivity (PC1_DTF) as a representative phenotypic trait. The PC1_DTF trait was derived from a multivariate PCA analysis on days to flowering and photoperiod sensitivity data collected in 10 different environments[13].

The Ka/Ks ratio was computed using KaKs calculator v2.0[69]. For each gene, the consensus sequence of each accession was extracted using bcftools consensus v1.10.2. The calculator compares the pangenome gene sequence with the gene sequence of each accession to identify non-synonymous and synonymous variants and then computes the ratio. The calculator reported NA when there were no variants in a specific accession or when the denominator of the Ka/Ks ratio was zero. It was possible to compute the analysis for 30,484 of 34,338 genes. Sometimes the length of one of the two compared sequences was not divisible by three so the sequence could not be read in triplets (Supplementary Data 4). The average Ka/Ks value per gene was used to assess the significance of the sample median (Supplementary Table 3).

$F_{ST}$ analysis involved the separate testing of PAVs in the Mesoamerican and Andean gene pools by comparing the frequency of each PAV between wild and domesticated forms. Each PAV was considered as a single locus (0/1) and $F_{ST}$ was calculated by applying the formula $F_{ST} = (H \, total - H \, within)/H \, total$, where H is the heterozygosity[70]. The same procedure was applied to wild accessions when comparing the Mesoamerican and Andean gene pools. Only PAVs in the top 5% of the $F_{ST}$ distribution were considered as candidates.

The functions of interesting PAVs and those associated with *A. thaliana* orthologs detected by OrthoFinder[61] were investigated manually in the NCBI database (https://www.ncbi.nlm.nih.gov/).

Phylogenetic analysis was conducted using bcftools[64], by first extracting SNPs from core genes and PAVs, followed by filtering. We applied the following filtering parameters: excluded insertions and deletions (--exclude-types indels), included only biallelic variants (--min-alleles 2 --max-alleles 2), included variants where the proportion of missing data was less than or equal to 0.5 (--include "F_MISSING ≤ 0.5"), excluded variants with minor allele frequency less than or equal to 0.01 (--exclude "MAF ≤ 0.01"), and excluded monomorphic sites that were homozygous for the reference (--min-ac 1). This process resulted in two final datasets: 1,451,663 SNPs for the core genes and 338,212 SNPs for the PAVs. The two filtered datasets were used to calculate the genetic distance between individuals and compute maximum composite likelihood values with 1000 bootstraps for the NJ tree in MEGA11[71]. The resulting trees were visualized in FigTree (http://tree.bio.ed.ac.uk/software/figtree/).

The filtered dataset of SNPs in core genes was also used to quantify the genetic diversity within each genetic group by estimating π. The --window-pi vcftools flag was used to obtain measures of nucleotide diversity in 250-kb windows. The windowed-pi estimates were then divided by the total number of SNPs to calculate a global estimate for each genetic group.

PAV-based Fisher's exact test with the false discovery rate corrected for multiple comparisons was applied in R to identify PAVs that differed significantly in frequency between the Mesoamerican and Andean gene pools for the American and European accessions.

The principal components related to days-to-flowering and photoperiod sensitivity (PC_DTF), derived from a multivariate PCA analysis on days to flowering and photoperiod sensitivity data collected in 10 different environments[13], were used for PAV-based GWAS using both the mixed linear model (MLM)[72] and the fixed and random model

circulating probability unification (FarmCPU) model[73] implemented in the R package GAPIT v3[74]. The threshold for each scan was determined by the Bonferroni corrected *p* value at $\alpha = 0.05$ ($p \leq 7.07E{-}06$). The kinship matrix (IBS method) calculated with Tassel 5[75] and the population structure at K2[13] were included in the models as fixed factors. Quantile–quantile (Q–Q) plots were obtained by plotting the observed −log10(*p* values) against the expected -log10(*p* values) under the null hypothesis of no association.

The spatial interpolation on wild common bean genotypes in relation to the number of PAVs per individual was performed using the gstat package in R. We applied ordinary Kriging to create an interpolation model. The geographic coordinates and number of PAVs were merged into a single dataset. A prediction grid was generated over the study area. The output GeoTIFF file was imported into QGIS for map visualization.

## Reporting summary
Further information on research design is available in the Nature Portfolio Reporting Summary linked to this article.

## Data availability
The 109 raw WGS reads generated in this study have been deposited in the National Center of Biotechnology Information (NCBI) Sequence Read Archive (SRA) under BioProject number PRJNA1042929. Additional WGS data, comprising 10 and 220 raw WGS reads, were sourced from BioProject numbers PRJNA910538 and PRJNA573595, respectively. The RNA-Seq data from this study have been deposited in the NCBI SRA under BioProject number PRJNA1042929. Additionally, 21 RNA samples were sourced from BioProject number PRJNA212729. The reference genome G19833 v2.1 is available on Phytozome at [https://phytozome-next.jgi.doe.gov/info/Pvulgaris_v2_1]. The other four high-quality genomes have been deposited in the NCBI SRA under BioProject number PRJNA1042929. The pan-genome assembly and its annotation have been deposited in Figshare [https://doi.org/10.6084/m9.figshare.24573874]. Source data are provided with this paper.

## Code availability
Custom codes used in this study have been deposited on GitHub [https://doi.org/10.5281/zenodo.12191159].

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

## Acknowledgements

This work was developed within the Horizon2020 Project INCREASE, grant agreement number 862862 (R.P., https://www.pulsesincrease.eu/), which aims to develop new tools and strategies for conserving food legume genetic resources and promoting their sustainable use. We thank the Department of Energy Joint Genome Institute and collaborator Prof. Phillip McClean for allowing us to incorporate the unpublished *Phaseolus vulgaris* v2.1 G19833 into our pan-genome analysis. We acknowledge the support provided by the BEAN ADAPT project (R.P.), founded through the ERA-CAPS Program, 2014 Call, Expanding the European Research Area in Molecular Plant Sciences. We acknowledge the support provided by the Italian Government, Miur, through the grant NextBEAN FIRB project RBFR13IDFM_001 (R.P.) and PARDOM PRIN project 20177RL4KL (R.P.). We acknowledge the support of the African Union Commission, within the project "Enhancing the nutrition and health of smallholder farmers in East Africa through increased productivity of biofortified common bean and improved postharvest handling", grant contract identification no. AURG II-2-087-2018 (R.P.). We acknowledge the support of the Agence Nationale de la Recherche, grant EGERI ANR-22-CE20-0022 (V.G.) as well as the support provided to IPS2 by Saclay Plant Sciences-SPS ANR-17-EUR-0007. We acknowledge support from the Australian Research Council grant no. DP200100762 (D.E.) and resources provided by the Pawsey Supercomputing Centre. We thank Prof. Simone Pesaresi and Dr. Giacomo Quattrini for the spatial interpolation support in preparing Fig. 5b. We also thank Scott Jackson and Maud Tenaillon for their valuable scientific discussions and suggestions.

## Author contributions

M.D., D.E., V.G., and R.P. conceived and managed the project. G.C., L.V., M.D., and R.P. wrote the article. G.C., L.V., R.A., J.I.M., P.E.B., L.R., G.F., G.L., A.P., A.B., E.Be., V.D.V., L.N., J.J.F.F., M.R., O.M.A., P.L.M., M.R., T.G., K.N., J.C.A.D., A.G., V.G., E.Bi., M.D., D.E., and R.P. contributed to the editing of the article. G.C. and L.V. contributed to the organization of the Supplementary Materials. L.V., G.M., M.R., M.D., and R.P. produced data concerning the MIDAS and G12873 genomes and the pan-genome development. J.C.A.D., A.G., C.K., and V.G. produced data concerning the BAT93 and JaloEPP558 genomes. G.C., L.V., A.B., and M.R. conducted data analysis. All authors read and approved the article.

## Competing interests

The authors declare no competing interests.

## Additional information

[1]Department of Agricultural, Food and Environmental Sciences, Marche Polytechnic University, 60131 Ancona, Italy. [2]Department of Biotechnology, University of Verona, 37134 Verona, Italy. [3]Centre for Applied Bioinformatics and School of Biological Sciences, University of Western Australia, Perth, WA 6009, Australia. [4]Department of Life Sciences and Biotechnology, University of Ferrara, 44100 Ferrara, Italy. [5]Regional Service for Agrofood Research and Development (SERIDA), Ctra AS-267 PK 19, 33300 Asturias, Spain. [6]Genartis s.r.l, 37126 Verona, Italy. [7]Institute of Biotechnology and Molecular Biology, UNLP-CONICET, CCT La Plata, La Plata, Argentina. [8]Department of Agronomy and Plant Genetics, University of Minnesota, St. Paul, MN 55108-6026, USA.

[9]Department of Agriculture, University of Sassari, 07100 Sassari, Italy. [10]CBV—Centro per la Conservazione e Valorizzazione della Biodiversità Vegetale, University of Sassari, 07041 Alghero, Italy. [11]School of Agricultural, Forestry, Food and Environmental Sciences, University of Basilicata, 85100 Potenza, Italy. [12]Leibniz Institute of Plant Genetics and Crop Plant Research (IPK), 06466 Seeland, Germany. [13]CNRS, INRAE, Institute of Plant Sciences Paris-Saclay (IPS2), University of Evry, University Paris-Saclay, 91405 Orsay, France. [14]INRAE, Genotoul Bioinformatics Platform, Applied Mathematics and Informatics of Toulouse, Sigenae, MIAT, UR875 Castanet Tolosan, France. [15]These authors contributed equally: Gaia Cortinovis, Leonardo Vincenzi. ✉e-mail: valerie.geffroy@universite-paris-saclay.fr; massimo.delledonne@univr.it; Dave.Edwards@uwa.edu.au; r.papa@univpm.it

