## [Peer Review File · Nature Communications]

Adaptive gene loss in the common bean pan-genome during range expansion and domesticationReviewers' Comments:

Reviewer #1:

Remarks to the Author:

The paper presents a comprehensive study of the common bean's pan-genome, using high-quality genomes and whole-genome sequencing data from various genotypes. Key findings include the identification of core and accessory genes, with the former associated with homeostasis and catabolic processes and the latter enriched in defense and response mechanisms. The study provides insights into the adaptive gene loss during expansion and domestication, and identifies distinct gene pools and subgroups. The research is significant for understanding the genetic diversity, genome dynamics, and adaptive evolution in beans.

Comments:

The discovery of significant gene loss, potentially adaptive in nature, aligns with models of evolution by genomic reduction and is a notable aspect of the paper. The GWAS conducted also provided intriguing results. It could be beneficial if the authors expand upon these GWAS findings in the main text and figures.

The work is significant in the field, particularly the projection of Presence/Absence Variations (PAVs) onto a large number of genomes. However, providing more details in the revised version would strengthen the reproducibility and impact. Details on pan-genome creation, particularly the assembly process from unmapped short reads, need to be clearer. Were individual accession's reads assembled or were pooled reads across samples assembled? The authors describe clustering but it's not immediately apparent how these were then assembled into contigs.

Pan-genome annotation is notoriously difficult, especially in the absence of data giving expression evidence (i.e. line specific RNAseq or IsoSeq across tissues, environments, etc). The possibility that PAVs reflect allelic variation rather than true gene content should be addressed. Acknowledgment of potential errors, such as annotation mistakes due to allelic variants, genes being split, etc, should be discussed.

A significant area of ambiguity in the paper pertains to the interpretation and usage of the term 'Presence-Absence Variation (PAV)'. It is unclear when the authors refer to an accession having a higher number of PAVs, as the term seems to be used in varying contexts. It appears that this might indicate an accession possessing a greater number of genes demonstrating PAV across the sampled genotypes. However, this inference is not explicitly stated, leading to potential misunderstandings. For instance, the phrase 'more PAV loss' in the abstract seems to imply a greater absence of PAV genes. This could be interpreted as a reduction in the presence-absence variation itself, but the paper does not explicitly clarify this point. Similarly, in Figure 2d, the 'Number of PAVs' is mentioned, but the exact meaning of this metric remains vague. To enhance clarity and precision, it is essential for the paper to define 'PAV' and what it means to have "more PAVs" explicitly and consistently. The term should be used in a manner that unmistakably conveys whether it refers to the presence or absence of specific genes, the number of variations in gene presence or absence, or other related interpretations. This improvement would greatly aid in the reader's understanding and interpretation of the study's findings.

Line 39: The term "gene loss as a key adaptive trait" could be more accurately described as a significant genetic change rather than a trait.

Line 57: Clarification on the "non-iterative approach" would help in understanding the methodology better.

Line 66: The use of "forms" in reference to domesticated varieties is ambiguous. Clarification on

whether this pertains to morphological differences or another aspect would be useful.

Figure 1b: The use of a log scale on the y-axis and the clarity of statistical comparisons need improvement.

Not sure what is meant in L357: "To ensure that the NRRs identified through this method didn't encompass orthologous genes already existing in the PV442 reference genome, we specifically employed highly conserved BUSCO genes." Could you elaborate?

Figure 3A: Enhancing clarity by indicating the number and nature of genes shown would improve the figure's utility.

Line 95: Consideration of alternative explanations for the observed PAV genes, such as neutrality, drift, or annotation issues, would provide a more complete view.

The latitude and longitude values in Figure 4B should be corrected.

Reviewer #2:

Remarks to the Author:

This manuscript reports the resequencing of a diverse collection of *Phaseolus vulgaris* accessions, from distinct populations in Mesoamerica, Peru/Ecuador, and the Andes, and a pangenome representing the union of sequences from this collection. The results indicate that gene loss is a consistent feature in the two bean main domestication events. Overall, I think this research looks to be solid and well conducted and described. I have found few corrections to recommend. My main suggestion is that I think the discussion could be fleshed out. There is a wealth of new information here - particularly rich due to the two distinct Andean and Mesoamerican populations and the associated independent domestications. Where there were PAVs in one population, how often were these seen (or not seen) in the other population? If the PAV patterns were different, it would be nice to see at least some speculation about the potential biological significance or reasons.

My other major comment is that some critical underlying data seems not to have been made available. Specifically: the genome assemblies and annotations should be available, as they must have been essential to this analysis and the results.

Details:

line 179: "may suggests"  "may suggest"

line 202 and subsequent: it is not entirely clear to me whether loss or gain of Phvul.003G265200 was selected for during domestication. From the context of the full paragraph, I think the loss was selected; but "PAV ... frequency declined" is ambiguous, because this is read as "presence-absence ... frequency declined" -- but which is it: the presence that declined in the domesticated population, or the absence that declined?

Further: what is the status of Phvul.003G265200 in the Andean populations (wild and domesticated)? For example, if loss is common in the Mesoamerican but not Andean domesticates, why would this be interpreted as a footprint of selection?

Related point regarding Phvul.003G265200: it would be helpful to see a discussion - even if speculative - about why loss of this gene might be beneficial. If the gene is "required for symbiotic interactions" in bean (which hasn't been established), then one would think that it would be beneficial in lines grown in an intensive agroecosystem where, presumably, nitrogen is limited.

Data availability:

I see that the "pan-genome assembly and its annotation" have been made available at FigShare. It looks to me like assemblies and annotations of the genomes (BAT93, JaloEEP558, MIDAS, G12873) have not been made available. The raw reads are insufficient.

Reviewer #3:

Remarks to the Author:

Cortinovis et al. present the first common bean pan-genome based on five-quality genomes and whole-genome reads representing 339 accessions. The study aimed to look for the non-reference sequences and genes. The authors also explored the PAVs, species range expansion, and adaptations. The topics addressed are interesting, but the manuscript has several problems as presented. Some of the analyses are not appropriate and also need to be clarified, and some of the conclusions lack evidence to support them. Several issues with the manuscript need to be addressed before the manuscript is published.

1. There are many issues with the methods. In the methods, since you have five assembled genomes with long reads, have you found some chromosome rearrangement and other structure variations? In the Methods, "Pan-genome annotation," lines 376-377, "Proteins from *P. vulgaris* and correlated species (*Medicago truncatula* and *Glycine soja*) pluscRNA-Seq data (unpublished data from [18]) were aligned to the genome and used as extrinsic evidence". Sorry, which RNAseq data? If you generated, please clarify the method; if you used the published data, please cite the literature.

About the "PAV calling," do you mean the PAV calling here is only focused on the gene present or absent? Since you use those 339 low-coverage WGS accessions to call SVs via Illumina reads, how accurate are those PAVs? You may lose some long PAVs and cause some bias in the analysis.

About the variant calling, have you done some filtering for the raw variants? How do you know all of the SNPs or indels are truly positive? What is the SFS for the SNPs and indels?

The author treated the nonsynonymous variants as deleterious and based on the Ka/Ks ratio to argue that "we observed a significant increase ($p = 0.048$) in the proportion of putative harmful variants among the rare genes compared to the soft-core genes (Fig. 1b; Supplementary Table 1g). These results may reflect the lower effective population size of the PAVs (reducing the efficiency of purifying selection) and/or the higher fitness gain from purging genes that have accumulated deleterious mutations (loss-of-function mutations). This is incorrect because some nonsynonymous variants could not be deleterious (see Kono et al., 2018 10.1534/g3.118.200563). I would run some pipelines (such as BAD_Mutation, Kono et al., 2018 10.1534/g3.118.200563) to predict the deleterious variant if the author wants to explore this topic further. And here, what do you mean by "the higher fitness gain"?

2. There are many issues without enough evidence to support them.

Line 108-110, "The analysis of variance conducted on M1/Jalisco-Durango accessions, considering the first component for the days to flowering, revealed that cluster A is significantly later-flowering than cluster B ". how did the authors know "the first component for the days to flowering"? Please show the evidence.

Line 105, "We constructed a neighbour-joining (NJ) phylogenetic tree (Fig. 2a)", I would add the "based on the SNPs in the core gene". So the PCA is only based on PAVs or the SNPs in the non-core genes (you also called it as PAVs, I would not use PAV refer to non-core gene because it is really confusing and it is hard for audience to know if this is the non-core genes or PAVs)?

Line 111-113, "This outcome also confirmed that our pan-genome enhances the characterization of genetic diversity and improves its analysis, exploitation, and management ."How did the author reach

this conclusion? Is that because the previous studies could not capture the genetic structure and diversity?

Line 127, "M1/Jalisco-Durango and A2/Peru races have more PAVs than the other subgroups in the same gene pool(Fig. 2d)." well, apparently, A2 is much lower than M1, M2, and WA. Why did the author reach this conclusion?

Figure 3a: I am confused about the reference and NRR. Do you mean for each accession's genome, you can divided them into reference and NRR parts. But if they are the same as the reference, why there are some absent there?

Line 167, 168, "with Mesoamerican accessions featuring more PAVs per accession than Andean ones (Fig. 3b, c; Supplementary Table 3a)". Based on Figure 3b, the average coverage of the M1, M2, and A1, A2, and A3 have less than 8 coverage. With such low coverage, it is hard to say the difference is real or due to the false positive of the PAV calling.

3. The authors used VCF tools to calculate π . Tools were designed to calculate the population genetic statistics for diploid samples. Still, samples in this study are highly selfing (which will be treated as haploid), and can not VCFtools to calculate π . I would suggest using I would use something like ANGSD (10.1186/s12859-014-0356-4)/angst-wrapper (10.1111/1755-0998.12578), which has haploid mode to calculate those population statistics.

4. All of the scripts and analytical tools used for a primarily computational project like the work reported here must be made available through Github, Bitbucket, or a similar version control repository. Because the scripts are essential to reproducing this work, they should be available to reviewers and readers. These resources are essential partly because bcftools and other similar utilities include many options that alter the resulting output.

Minor:

1. Line 270, the citation for Phytozome should also include this paper "10.1093/nar/gkr944".

POINT-BY-POINT RESPONSE TO THE REVIEWERS' COMMENTS

FOR ALL REVIEWERS: In addition to the changes made following your comments, we have rectified a typo related to the classification of core genes and PAVs. Specifically, we removed 590 genes from the total count of PAVs. This is because we erroneously classified them simultaneously as PAVs and core genes, instead of classifying them as core genes only. These adjustments do not alter the overall interpretation of our results but rather reinforce them. These changes are also visible in the track changes version of the manuscript.

FOR ALL REVIEWERS: The data are under embargo until publication. Temporary access to the data availability can be found at the following links:

- NCBI
<https://dataview.ncbi.nlm.nih.gov/object/PRJNA1042929?reviewer=o2s2gfk4k9tesdgohqvs7t51lm>
- FIGSHARE
<https://figshare.com/s/56c9d8cc36c7ea3c195d>

Reviewer #1 (Remarks to the Author):

The paper presents a comprehensive study of the common bean's pan-genome, using high-quality genomes and whole-genome sequencing data from various genotypes. Key findings include the identification of core and accessory genes, with the former associated with homeostasis and catabolic processes and the latter enriched in defense and response mechanisms. The study provides insights into the adaptive gene loss during expansion and domestication, and identifies distinct gene pools and subgroups. The research is significant for understanding the genetic diversity, genome dynamics, and adaptive evolution in beans.

Comment: The discovery of significant gene loss, potentially adaptive in nature, aligns with models of evolution by genomic reduction and is a notable aspect of the paper. The GWAS conducted also provided intriguing results. It could be beneficial if the authors expand upon these GWAS findings in the main text and figures.

Response: We followed the suggestion and extended this section within the main text and with a new figure (Fig. 3).

Comment: The work is significant in the field, particularly the projection of Presence/Absence Variations (PAVs) onto a large number of genomes. However, providing more details in the revised version would strengthen the reproducibility and impact. Details on pan-genome creation, particularly the assembly process from unmapped short reads, need to be clearer. Were individual accession's reads assembled or were pooled reads across samples assembled? The authors describe clustering but it's not immediately apparent how these were then assembled into contigs.

Response: To address your specific question, the Illumina data from 339 accessions were aligned to the preliminary pan-genome. Then, the unmapped reads were pooled and *de novo* assembled. The resulting contigs were then incorporated into the final pan-genome assembly.

We revised the part entitled 'Pan-genome construction' within the Methods section to provide a more accurate description of the pan-genome creation process.

Comment: Pan-genome annotation is notoriously difficult, especially in the absence of data giving expression evidence (i.e. line specific RNAseq or IsoSeq across tissues, environments, etc). The possibility that PAVs reflect allelic variation rather than true gene content should be addressed. Acknowledgment of potential errors, such as annotation mistakes due to allelic variants, genes being split, etc, should be discussed.

Response: To address these concerns, we employed a hybrid approach for pan-genome annotation by combining *ab initio* prediction with extrinsic supporting evidence. This evidence included *P. vulgaris* RNA-Seq data as well as protein sequences of *P. vulgaris* and related species (*Medicago truncatula* and *Glycine soja*). It is important to note that the RNA-Seq data include MIDAS, for which we have a high-quality genome. This approach leverages both transcriptomic and proteomic information, enhancing the accuracy of gene prediction compared to using computational methods alone. Thus, we are confident that through our procedure we minimized potential annotation mistakes. We revised the part entitled 'Pan-genome annotation' in the Methods section for clarity, as well as added RNA-Seq data information in both the Methods and Data Availability sections. Moreover, we validated the accuracy of the detected NRRs through a comparative analysis using highly conserved BUSCO genes. Our analysis confirmed that the identified NRRs reflected gene content rather than allelic variation. Specifically, we examined the entire set of 4,947 MIDAS and 4,812 G12873 BUSCO genes shared with the reference genome G19833 v2.1 within the NRRs using BLASTp, and the outcome revealed that only a minimal number of genes (i.e., 7 in MIDAS and 37 in G12873) were identifiable within the NRRs. We revised the part entitled 'Pan-genome construction' within the Methods section to include this information.

Comment: A significant area of ambiguity in the paper pertains to the interpretation and usage of the term 'Presence-Absence Variation (PAV)'. It is unclear when the authors refer to an accession having a higher number of PAVs, as the term seems to be used in varying contexts. It appears that this might indicate an accession possessing a greater number of genes demonstrating PAV across the sampled genotypes. However, this inference is not explicitly stated, leading to potential misunderstandings. For instance, the phrase 'more PAV loss' in the abstract seems to imply a greater absence of PAV genes. This could be interpreted as a reduction in the presence-absence variation itself, but the paper does not explicitly clarify this point. Similarly, in Figure 2d, the 'Number of PAVs' is mentioned, but the exact meaning of this metric remains vague. To enhance clarity and precision, it is essential for the paper to define 'PAV' and what it means to have "more PAVs" explicitly and consistently. The term should be used in a manner that unmistakably conveys whether it refers to the presence or absence of specific genes, the number of variations in gene presence or absence, or other related interpretations. This improvement would greatly aid in the reader's understanding and interpretation of the study's findings.

Response: We agree that clarity and precision in using the term presence-absence variations (PAVs) is crucial for ensuring reader understanding and interpretation of our findings. We have carefully considered your suggestions and revised the manuscript accordingly to avoid potential misunderstandings. Specifically, we defined PAVs as genes that are present in some individuals while absent in others. When referring to an accession or population having a "higher number of PAVs" we now explicitly indicate that this means a greater number of genes (presences) across the sampled genotypes. We also clarified this in Figure 2d and Figure 4. Additionally, we modified the phrase "more PAV loss" in the abstract to "fewer PAVs".

Comment: Line 39: The term "gene loss as a key adaptive trait" could be more accurately described as a significant genetic change rather than a trait.

Response: We agree and have modified the manuscript as suggested.

Comment: Line 57: Clarification on the "non-iterative approach" would help in understanding the methodology better.

Response: With the definition 'non-iterative approach' we refer to a paired genome alignment strategy through the independent alignment of the reference genome to the other high-quality genomes. To improve clarity, we enhanced the detailed explanation of this procedure in the part entitled 'Pan-genome construction' of Methods section.

Comment: Line 66: The use of "forms" in reference to domesticated varieties is ambiguous. Clarification on whether this pertains to morphological differences or another aspect would be useful.

Response: In the context of our study, the term 'forms' denotes the biological status of the common bean accessions. These forms can be broadly categorized into two main groups: wild and domesticated. Wild forms, which have not been subjected to human cultivation or selection, typically exhibit a wide range of morphological diversity, reflecting adaptation to various environmental conditions. Domesticated forms include: I) Landraces, local and dynamic varieties that have evolved through natural and artificial selection over millennia adapting to specific agro-environments by traditional farming practices. II) Cultivated varieties that have been managed by humans for desirable traits such as yield, taste, size, and resistance to pests and diseases. Domesticated forms often exhibit significant morphological and genetic differences compared to their wild counterparts, as they have been subjected to natural and artificial selection over many generations. Therefore, when we use the term 'forms' in this study, we are specifically referring to the differentiation between these two broad categories. This differentiation encompasses not only visible differences in appearance (morphological disparities) but also underlying genetic variations that have arisen as a result of the domestication and diversification processes. These genetic variations can include changes in gene sequences, gene expression patterns, and genomic structure, all of which contribute to the distinctiveness of domesticated accessions compared to their wild ancestors. We have included this information in Supplementary Table 5b (Column E – Biological Status).

Comment: Figure 1b: The use of a log scale on the y-axis and the clarity of statistical comparisons need improvement.

Response: We addressed this feedback by adding the relevant statistical information to the figure caption. Additionally, Supplementary Table 1g was modified for clarity.

Comment: Not sure what is meant in L357: “To ensure that the NRRs identified through this method didn't encompass orthologous genes already existing in the PV442 reference genome, we specifically employed highly conserved BUSCO genes.” Could you elaborate?

Response: As reported above, we validated the accuracy of the detected NRRs through a comparative analysis using highly conserved BUSCO genes. Our analysis confirmed that the identified NRRs reflected gene content rather than allelic variation. Specifically, we examined the entire set of 4,947 MIDAS and 4,812 G12873 BUSCO genes shared with the reference genome G19833 v2.1 within the NRRs using BLASTp, and the outcome revealed that only a minimal number of genes (i.e., 7 in MIDAS and 37 in G12873) were identifiable within the NRRs. We revised the part entitled ‘Pan-genome construction’ within the Methods section to include this information. We revised and reformulated this part in accordance with your suggestion.

Comment: Figure 3A: Enhancing clarity by indicating the number and nature of genes shown would improve the figure's utility.

Response: We modified the figure and its caption (Fig. 4a) to include the information required.

Comment: Line 95: Consideration of alternative explanations for the observed PAV genes, such as neutrality, drift, or annotation issues, would provide a more complete view.

Response: We tested alternative explanations in other manuscript sections, specifically in the paragraphs titled 'Pan-genome shrinkage during wild expansion to South America' and 'Footprints of selection for gene loss during domestication.' In these sections, we conducted analyses and provided evidence supporting the role of natural selection in shaping PAV diversity, particularly during common bean wild expansion and domestication. Our findings suggest that natural selection drives gene loss over mere stochastic responses and highlight that these genes are potentially involved in key adaptive traits. Thus, since gene loss is considered functionally equivalent to other loss-of-function mutations, our observation that PAVs include a higher K_a/K_s ratio, compared to the core genes, could be due to a higher fitness gain from purging genes that have accumulated non-synonymous mutations (loss-of-function mutations).

Comment: The latitude and longitude values in Figure 4B should be corrected.

Response: We revised the figure (Fig. 5b) in accordance with the suggestion. Additionally, we clarified the system used to indicate latitude and longitude values in the caption of the figure.

Reviewer #1 (Remarks on code availability):

Comment: Link <https://doi.org/10.6084/m9.figshare.24573874> does not seem to work for me.

Response: The link provided in the main text (<http://doi.org/10.6084/m9.figshare.24573874>) is currently inactive as the data are under embargo until publication. However, reviewers can still access the required data through the following temporary link: <https://figshare.com/s/56c9d8cc36c7ea3c195d>. This repository

contains the assemblies of the high-quality genomes, as well as the pan-genome, along with their annotations.

Reviewer #2 (Remarks to the Author):

This manuscript reports the resequencing of a diverse collection of *Phaseolus vulgaris* accessions, from distinct populations in Mesoamerica, Peru/Ecuador, and the Andes, and a pangenome representing the union of sequences from this collection. The results indicate that gene loss is a consistent feature in the two bean main domestication events. Overall, I think this research looks to be solid and well conducted and described. I have found few corrections to recommend.

Comment: My main suggestion is that I think the discussion could be fleshed out. There is a wealth of new information here - particularly rich due to the two distinct Andean and Mesoamerican populations and the associated independent domestications. Where there were PAVs in one population, how often were these seen (or not seen) in the other population? If the PAV patterns were different, it would be nice to see at least some speculation about the potential biological significance or reasons.

Response: We think that the comparison between Mesoamerican and Andean populations has been explored from various perspectives. In the paragraph titled 'Evolutionary trajectory of the common bean,' we investigated gene-pool differentiation using PAV-based Fisher's exact test between the domesticated Mesoamerican and Andean populations. Our analysis revealed that more than 60% of the PAVs exhibit statistically significant differences in frequency between the two populations. Notably, we identified 721 PAVs as diagnostic, indicating that they are present in one population with a frequency of 1 and completely absent in the other. We conducted GO enrichment analysis on these diagnostic PAVs and observed enrichment in processes related to metabolism, detoxification, and responses to stimuli. In the subsequent paragraph titled 'Pan-genome shrinkage during wild expansion to South America,' we applied *F_{st}* analysis between wild Mesoamerican and Andean populations, and investigated gene functions of candidate genes. We found enrichment for genes involved in pollen germination, innate immunity, abiotic stress tolerance, and root hair growth. Moving forward to the paragraph titled 'Footprints of selection for gene loss during domestication,' we performed *F_{st}* analysis between wild and domesticated populations, separately for each gene pool. Here we found 29 genes putatively under selection in common between the Mesoamerican and Andean populations, suggesting a potential phenomenon of convergent evolution. Interestingly, gene function investigation revealed that these genes are mainly involved in the tryptophan pathway, a compound crucial in several adaptive processes. We adjusted these three paragraphs to enhance clarity and argumentation, including rephrasing certain sentences for smoother readability. Additionally, we extended the investigation of the candidate Phvul.003G185200, identified by PAV-based GWA analysis, to provide further insights. We found a divergent distribution of this PAV between the two gene pools: Phvul.003G185200 is present in all Mesoamerican genotypes but only in 18% of Andean ones. Interestingly, genotypes lacking Phvul.003G185200 exhibit early flowering compared to those that carry it. We also included a new figure in support of this comparison (Fig. 3).

Comment: My other major comment is that some critical underlying data seems not to have been made available. Specifically: the genome assemblies and annotations should be available, as they must have been essential to this analysis and the results.

Response: We added the required data. You can access the updated data at the following link: <https://figshare.com/s/56c9d8cc36c7ea3c195d>. This repository contains the assemblies of the high-quality genomes, as well as the pan-genome, along with their annotations.

Comment: Details: line 179: "may suggests"  "may suggest"

Response: We corrected the text as suggested.

Comment: line 202 and subsequent: it is not entirely clear to me whether loss or gain of Phvul.003G265200 was selected for during domestication. From the context of the full paragraph, I think the loss was selected; but "PAV ... frequency declined" is ambiguous, because this is read as "presence-absence ... frequency declined" -- but which is it: the presence that declined in the domesticated population, or the absence that declined?

Response: We acknowledge the importance of clarifying the term 'PAV ... frequency declined'. The term 'presence/absence' is expressed in binary terms of 0/1, where '0' represents absence and '1' represents presence, specifically referring to a certain gene in a certain genotype. When considering a population, instead of a single genotype, the presence of a certain gene can be seen as the sum of the genotypes showing presence divided by all the population genotypes, resulting in a frequency value. These gene frequencies were instrumental in categorizing pan-genes into core or PAVs, and further in splitting the PAVs into sub-categories like soft-core, accessory, and rare (See paragraph titled 'Characterization of the common bean pan-genome'). This is why we discussed frequency in relation to presence/absence variations. To prevent misunderstandings, we revised the text to specify that the decline observed pertains to the presence of Phvul.003G265200 in the domesticated population. Corresponding modifications were made to the subsequent part and conclusion to ensure consistency. Additionally, we adjusted the y-axis of Fig. 6a and Fig. 6b from 'frequency' to 'genes' for clarity.

Comment: Further: what is the status of Phvul.003G265200 in the Andean populations (wild and domesticated)? For example, if loss is common in the Mesoamerican but not Andean domesticates, why would this be interpreted as a footprint of selection?

Response: Phvul.003G265200 is a putative PAV under selection for the Mesoamerican gene pool, whose presence declined by more than 60% during progression from the wild (0.94) to the domesticated (0.25) population. On the other hand, Phvul.003G265200 is present in equal frequency (1.00) in both wild and domesticated Andean populations, and consequently it is not putatively under selection during the Andean domestication. The fact that this PAV shows a footprint of selection only in the Mesoamerican population agrees with the parallel and independent domestications that these two gene pools experienced. Thus, the variations in the presence/absence of Phvul.003G265200 could represent a distinctive hallmark of the independent domestication processes between Mesoamerican and Andean common bean populations.

Comment: Related point regarding Phvul.003G265200: it would be helpful to see a discussion - even if speculative - about why loss of this gene might be beneficial. If the gene is "required for symbiotic interactions" in bean (which hasn't been established), then one would think that it would be beneficial in lines grown in an intensive agroecosystem where, presumably, nitrogen is limited.

Response: Common bean populations interact with various types of symbionts, indicating a complex relationship between host plants and specific strains of nitrogen-fixing bacteria, primarily from the genus *Rhizobium*. A speculative explanation for the biological importance of the loss of Phvul.003G265200 in

Mesoamerican domesticated genotypes could be related to specificity. Different genotypes of common bean preferentially associate with specific strains or species of rhizobia. Thus, it's plausible that the loss of this gene in domesticated genotypes enhances the flexibility of symbiotic interactions, allowing common bean populations to adapt to diverse environmental conditions and various symbiont partners. This speculation parallels the concept of the cost-benefit trade-off seen in resistance genes. Just as resistance genes often come with fitness costs, meaning that plants investing resources into defense mechanisms might have reduced growth or reproductive success compared to plants without these genes, the loss of Phvul.003G265200 may enable common bean populations to broaden their range of symbiotic interactions, allowing them to establish relationships with a greater diversity of symbionts instead of being limited to specific strains. However, further research and experimental validation would be necessary to conclusively elucidate the role of Phvul.003G265200 in common bean symbiotic interactions. We included this speculative hypothesis in the main text as recommended.

Comment: Data availability: I see that the "pan-genome assembly and its annotation" have been made available at FigShare. It looks to me like assemblies and annotations of the genomes (BAT93, JaloEEP558, MIDAS, G12873) have not been made available. The raw reads are insufficient.

Response: We added the required data. You can access the updated data at the following link: <https://figshare.com/s/56c9d8cc36c7ea3c195d>. This repository contains the assemblies of the high-quality genomes, as well as the pan-genome, along with their annotations.

Reviewer #2 (Remarks on code availability):

Comment: I have not reviewed the code. However, I have reviewed the provided data. See comments above regarding data availability.

Response: Custom codes used in this study are available at the following link on GitHub: https://github.com/PapaLab/Common_bean_pan_genome.git

Reviewer #3 (Remarks to the Author):

Cortinovis et al. present the first common bean pan-genome based on five-quality genomes and whole-genome reads representing 339 accessions. The study aimed to look for the non-reference sequences and genes. The authors also explored the PAVs, species range expansion, and adaptations. The topics addressed are interesting, but the manuscript has several problems as presented. Some of the analyses are not appropriate and also need to be clarified, and some of the conclusions lack evidence to support them. Several issues with the manuscript need to be addressed before the manuscript is published.

There are many issues with the methods:

Comment: In the methods, since you have five assembled genomes with long reads, have you found some chromosome rearrangement and other structure variations?

Response: We acknowledge that chromosome rearrangements were not investigated in our current study as they fall outside the scope of this manuscript. Our main focus was to construct a linear pangenome, with both wild and domesticated forms, representing the entire genetic diversity present in the common bean, and use it as a new reference for evolutionary analyses based on presence/absence variations (PAVs). In future investigations, we aim to explore other structural variations, including inter/intrachromosomal rearrangements, copy number variation, inversions, and transversions by transitioning to a graph-based pangenome approach.

Comment: In the Methods, "Pan-genome annotation," lines 376-377, "Proteins from *P. vulgaris* and correlated species (*Medicago truncatula* and *Glycine soja*) pluscRNA-Seq data (unpublished data from [18]) were aligned to the genome and used as extrinsic evidence". Sorry, which RNAseq data? If you generated, please clarify the method; if you used the published data, please cite the literature.

Response: We corrected the information related to the RNA-Seq data and provided detailed information in both the Methods and Data Availability sections. Specifically, we used RNA-Seq data from Bellucci et al. [19] and supplemented it with data from our own study. RNA-Seq data sourced from Bellucci et al., [19] are available at the BioProject number PRJNA212729, while the RNA-Seq data generated in this work are accessible via the following temporary link: <https://dataview.ncbi.nlm.nih.gov/object/PRJNA1042929?reviewer=o2s2gfk4k9tesdgohqvs7t51lm>.

Comment: About the "PAV calling," do you mean the PAV calling here is only focused on the gene present or absent? Since you use those 339 low-coverage WGS accessions to call SVs via Illumina reads, how accurate are those PAVs? You may lose some long PAVs and cause some bias in the analysis.

Response: Various thresholds have been applied in the literature to define gene presence, ranging from CDS coverage over 0.95 to exon coverage over 0.05 (<https://doi.org/10.1038/s41586-018-0063-9>; <https://doi.org/10.1111/nph.15413>; <https://doi.org/10.1038/nature12221>; <https://doi.org/10.1111/tpj.13515>, <https://doi.org/10.1038/ncomms13390>). In our study, we developed a specific threshold for PAV calling, termed the MIN threshold, as an alternative to the commonly used 0.05 threshold based on gene coverage. The MIN threshold is based on the minimum coverage value of 1000 randomly selected BUSCO genes (ORFs) for each accession, allowing for the definition of an accession-specific

threshold for calling genes as present. This approach was implemented to improve accuracy by mitigating the impact of sequencing depth, especially in large gene families. By adopting the MIN threshold, we minimized the risk of false positives in PAV calling and ensured the reliability of our results. We revised the part entitled 'PAV calling' within the Methods section to provide further clarification on this approach.

Comment: About the variant calling, have you done some filtering for the raw variants? How do you know all of the SNPs or indels are truly positive? What is the SFS for the SNPs and indels?

Response: In response to your comment, we have included an explanation of the filtering procedures applied to the raw variants in the 'Variant calling' and 'Data analysis' sections of the Methods. Specifically, the raw variants were filtered based on a minimum mapping quality (-q) of 20, consistent with Jayakodi et al. [47]. Additionally, for data analysis, we applied further filtering criteria, including the exclusion of insertions and deletions, retention of only biallelic variants, incorporation of variants with a proportion of missing data less than or equal to 0.5, exclusion of variants with a minor allele frequency less than or equal to 0.01, and removal of monomorphic sites. These stringent filtering steps were implemented to mitigate the detection of false positives, and align with other studies (<https://doi.org/10.1101/2023.08.31.555682>, <https://doi.org/10.1038/s41598-021-82437-4>, <https://doi.org/10.1038/s41467-023-37332-z>).

Following your question about the site frequency spectrum, we calculated the SFS for the SNPs by splitting the genotypes in the Mesoamerican and Andean populations. As shown in the figures below, the SFS exhibit a normal shape for both the gene pools, that can be compared to what is present in the literature on common bean (<https://doi.org/10.1371/journal.pone.0189597>), indicating that there are no significant anomalies. This analysis provides additional validation of the reliability of our SNP calling procedure and reinforces the robustness of our findings.

Comment: The author treated the nonsynonymous variants as deleterious and based on the Ka/Ks ratio to argue that "we observed a significant increase ($p = 0.048$) in the proportion of putative harmful variants among the rare genes compared to the soft-core genes (Fig. 1b; Supplementary Table 1g). These results may reflect the lower effective population size of the PAVs (reducing the efficiency of purifying selection) and/or the higher fitness gain from purging genes that have accumulated deleterious mutations (loss-of-function mutations). This is incorrect because some nonsynonymous variants could not be deleterious (see Kono et al., 2018 10.1534/g3.118.200563). I would run some pipelines (such as BAD_Mutation, Kono et al., 2018 10.1534/g3.118.200563) to predict the deleterious variant if the author wants to explore this topic further. And here, what do you mean by "the higher fitness gain"?

Response: We agree that the term "deleterious" may lead to misunderstandings, so we adjusted the main text accordingly. Specifically, we replaced "deleterious" with the more general term "non-synonymous" to better reflect the nature of our hypothesis and interpretation. Indeed, it is important to note that when we use the term "deleterious," we do not intend to imply a negative connotation, such as a loss of fitness, but rather to signify the loss of functional integrity. Thus, predicting deleterious variants, understood as disadvantageous mutations, is not our main focus here. Following these considerations, "the higher fitness gain" refers to the potential advantage gained by losing genes that have accumulated non-synonymous mutations (i.e., loss-of-function mutations). This interpretation aligns with our results on common bean. Indeed, our findings suggest that natural selection drives gene loss over mere stochastic responses, and highlight that these genes are potentially involved in key adaptive traits. Since gene loss is considered functionally equivalent to other loss-of-function mutations, our observation that PAVs include a higher Ka/Ks compared to the core genes, it could stem from an increased fitness advantage gained by purging genes that have accumulated non-synonymous mutations.

Comment: There are many issues without enough evidence to support them: Line 108-110, "The analysis of variance conducted on M1/Jalisco-Durango accessions, considering the first component for the days to flowering, revealed that cluster A is significantly later-flowering than cluster B ". how did the authors know "the first component for the days to flowering"? Please show the evidence.

Response: The first component for the days to flowering (PC1_DTF) trait, derived from a multivariate principal component analysis (PCA) on days-to-flowering and photoperiod sensitivity data collected in 10 different environments, was sourced from Bellucci et al. [13]. Detailed information on obtaining these data is described in the section titled 'Supplementary Note 2: Phenotyping' of the aforementioned paper. We have now explicitly provided this information and cited Bellucci et al. [13] throughout the manuscript, including in the main text, in the caption of Fig. 2c, and in the Methods section.

Comment: Line 105, "We constructed a neighbour-joining (NJ) phylogenetic tree (Fig. 2a)", I would add the "based on the SNPs in the core gene". So the PCA is only based on PAVs or the SNPs in the non-core genes (you also called it as PAVs, I would not use PAV refer to non-core gene because it is really confusing and it is hard for audience to know if this is the non-core genes or PAVs)?

Response: With the sentence "We constructed a neighbour-joining (NJ) phylogenetic tree (Fig. 2a)" we refer to the NJ trees performed with both SNPs located on core genes and SNPs located on PAVs. We modified this sentence and added this information in the parentheses referring to Supplementary Fig. 2a in addition to Fig. 2a. Regarding the second question, we will provide some clarifications. We classified the pan-genes into two main categories: core genes and PAVs. The core genes are present in all the genotypes, while the PAVs are partially shared among the genotypes or private to a single one (See paragraph titled 'Characterization of the common bean pan-genome' for more details). After the classification of the genes, in the paragraph titled 'Evolutionary trajectory of the common bean' we used the PAVs data in several analyses for exploring the diversity, structure, and evolutionary history of the common bean: PAV-based PCA, PAV-based GWAS, number of PAVs per genetic group, PAV-based Fisher's Exact test, Go enrichment analysis on candidate PAVs, etc... However, for phylogenetic reconstruction, where the fewer genetic gaps between genotypes the better the resolution of the analysis, the use of PAVs could create biases arising from the absences among the compared accessions. Thus, we tested this hypothesis by using SNPs: SNPs were divided into those located on core genes and those located on PAVs, and the two datasets were used to construct the NJ trees. Even though the general common bean population structure is maintained, our analysis confirmed the more suitability of core genes than PAVs for phylogenetic reconstruction since the NJ tree based on core SNPs properly grouped the wild Phl accession close to the wild Mesoamerican genotypes originating from Guatemala and Costa Rica (Fig. 2a), which are most closely related to the Phl gene pool.

Comment: Line 111-113, "This outcome also confirmed that our pan-genome enhances the characterization of genetic diversity and improves its analysis, exploitation, and management ." How did the author reach this conclusion? Is that because the previous studies could not capture the genetic structure and diversity?

Response: Yes, this conclusion is supported by the fact that our study genetically distinguished the M1/Durango-Jalisco races in relation to a key adaptive trait (flowering time) for the first time. Previous studies may not have achieved this resolution due to limitations associated with using a centric approach, such as

relying on a single reference genome. By leveraging a pan-genomic system, we were able to explore a broader spectrum of genetic variations and uncover previously unseen genetic structure within the M1/Durango-Jalisco races. Furthermore, this conclusion is reinforced by other parallel results; for instance, the PAV-based GWA analysis detected a high percentage of candidate genes related to flowering time that are located on non-reference regions (NRRs), indicating that the use of the pan-genome as a reference in our study was crucial for capturing a greater level of resolution in the genetic diversity present in the common bean species. We rephrased the main text for clarity.

Comment: Line 127, "M1/Jalisco-Durango and A2/Peru races have more PAVs than the other subgroups in the same gene pool(Fig. 2d)." well, apparently, A2 is much lower than M1, M2, and WA. Why did the author reach this conclusion?

Response: We agree that the statement "M1/Jalisco-Durango and A2/Peru races have more PAVs than the other subgroups in the same gene pool" is ambiguous. We rephrased the text related to this part for clarity. Here, we are specifically referring to the comparison among domesticated subgroups within each gene pool, namely, M1 vs M2 for the Mesoamerican gene pool and A1 vs A2 vs A3 for the Andean gene pool. Specifically, regarding the Mesoamerican gene pool, the M1/Jalisco-Durango subgroup shows more PAVs than the subgroup M2/Mesoamerican. Regarding the Andean gene pool, the A2/Peru race has more PAVs than the A1/Nueva Granada and A3/Chile races (Fig. 2d). These results are also supported by nucleotide diversity analysis (Fig. 2e). The observation that M1 and A2 exhibit more PAVs than the other domesticated subgroups belonging to the same gene pool aligns with a recent hypothesis. This hypothesis proposes that the M1/Durango-Jalisco and A2/Peru races were the first domesticated Mesoamerican and Andean populations, respectively, from which the M2, A1, and A3 races arose during a secondary domestication phase.

Comment: Figure 3a: I am confused about the reference and NRR. Do you mean for each accession's genome, you can divide them into reference and NRR parts. But if they are the same as the reference, why there are some absent there

Response: The figure represents the 0/1 matrix illustrating presence and absence variations (PAVs) across a representative panel of 99 common bean accessions, specifically selected for data analysis from the larger set of 339 accessions used in pan-genome development. To elucidate the heatmap representation, it's essential to understand the methodology behind our pan-genome construction and PAV calling process. The term "reference" refers to the initial reference genome used as a baseline for comparison, while "NRRs" encompass regions absent in this reference but present in other genotypes within our pan-genome dataset. As detailed in the paragraph titled 'Characterization of the common bean pan-genome' and the corresponding Methods section, we constructed the pan-genome by independently aligning the reference genome with other high-quality genomes. Regions absent from the reference were then extracted and incorporated into the pan-genome. Additionally, NRRs were identified by aligning the short-read WGS data from 339 representative common bean accessions to the reference genomes, and these regions were also integrated into the pan-genome. This process resulted in a comprehensive pan-genome composed of the reference genome and additional sequences absent in the reference but present in other genotypes (NRRs) (See Supplementary Table 1a for details). Subsequently, we used the common bean pan-genome as the new reference for mapping short-read WGS data from 339 representative accessions and performing PAVs calling. Specifically, genes were categorized based on their frequency of occurrence: those present in all genotypes

were classified as core genes, while partially shared or genotype-specific genes were classified as PAVs (See Supplementary Table 1b). Consequently, the presence of absences on the reference genome in the figure (Fig. 4a) refers to variants identified in other genotypes within our pan-genome dataset, which are not present in the reference genome.

Comment: Line 167, 168, "with Mesoamerican accessions featuring more PAVs per accession than Andean ones (Fig. 3b, c; Supplementary Table 3a)". Based on Figure 3b, the average coverage of the M1, M2, and A1, A2, and A3 have less than 8 coverage. With such low coverage, it is hard to say the difference is real or due to the false positive of the PAV calling.

Response: See comment number 3.

Comment: The authors used VCF tools to calculate pi. Tools were designed to calculate the population genetic statistics for diploid samples. Still, samples in this study are highly selfing (which will be treated as haploid), and can not VCFtools to calculate pi. I would suggest using I would use something like ANGSD (10.1186/s12859-014-0356-4)/angst-wrapper (10.1111/1755-0998.12578), which has haploid mode to calculate those population statistics.

Response: While the common bean is highly selfing, it remains a diploid species. Thus, while it may have regions of homozygosity due to selfing, it also retains heterozygous regions in its genome. Indeed, the genome of each accession is mainly composed of homozygous regions, but it also includes a non-negligible number of heterozygous regions which should not be ignored. Therefore, treating the common bean solely as haploid may not accurately reflect its genetic structure. Moreover, the same approach was also used in other works (<https://www.nature.com/articles/s41598-023-39399-6>; <https://www.nature.com/articles/s41467-023-37332-z>; <https://www.mdpi.com/2223-7747/9/9/1238>), where authors successfully used VCFtools to estimate various population genetics statistics (including pi) in common bean. We appreciate your suggestion to use haploid-mode tools like ANGSD/angst-wrapper for calculating population statistics. However, for our specific analysis, we believe that treating the common bean as a diploid species ensures a more realistic representation of its genetic structure and diversity.

Comment: All of the scripts and analytical tools used for a primarily computational project like the work reported here must be made available through Github, Bitbucket, or a similar version control repository. Because the scripts are essential to reproducing this work, they should be available to reviewers and readers. These resources are essential partly because bcftools and other similar utilities include many options that alter the resulting output.

Response: We completely agree with the importance of ensuring data reproducibility. To address this concern, we provided the two custom codes for identifying functional domains and calculating the pan-gene and core gene sizes, respectively. These codes are accessible via the following GitHub link: https://github.com/PapaLab/Common_bean_pan_genome.git. Additionally, we have included comprehensive details about the software, packages, and parameters for each analysis employed in the Methods section. We are confident that, by combining these efforts, we have ensured the accessibility and reliability of our findings.

Comment: Minor: Line 270, the citation for Phytozome should also include this paper "10.1093/nar/gkr944".

Response: We added the suggested citation for Phytozome to the bibliography. It is listed as reference [38].

Reviewers' Comments:

Reviewer #1:

Remarks to the Author:

The authors have addressed all of my comments.

Reviewer #2:

Remarks to the Author:

The authors have submitted a thoughtful, careful revision that satisfactorily addresses my comments from the initial manuscript submission.

Reviewer #3:

Remarks to the Author:

All of my comments have been addressed. I have no further comments.

POINT-BY-POINT RESPONSE TO THE REVIEWERS' COMMENTS

We are pleased to note that all reviewers have indicated that their comments have been satisfactorily addressed.

Reviewer #1

Reviewer #1 (Remarks to the Author):

The authors have addressed all of my comments.

Reviewer #1 (Remarks on code availability):

The code covers some aspects of the paper, no installation required.

Reviewer #2

Reviewer #2 (Remarks to the Author):

The authors have submitted a thoughtful, careful revision that satisfactorily addresses my comments from the initial manuscript submission.

Reviewer #3

Reviewer #3 (Remarks to the Author):

All of my comments have been addressed. I have no further comments.